# Polarization manipulation of electromagnetic interference shielding effectiveness utilizing graphene film-based metamaterials

Zhe Wang [1], Haoran Zu [1,2] ✉, Zhi Luo[1], Yu Zhou[1], Yiping Ren[1], Huazhang Zhang [1], Zixin Zhang[1], Wei Qian[1], Huaqiang Fu[1], Lun Li[1], Hao Feng[1], Pengfei Chen[1], Long Zhang[1], Hao Yuan[1], Junkang Xia[1], Xin Zhao[1], Shuxin Li [1] ✉ & Daping He [1] ✉

The rapid advancement of the Information Age needs the development of intelligent electromagnetic interference shielding materials that respond to real-time environmental alterations. Herein, we illuminate the electromagnetic interference shielding mechanisms of carbon-based materials and verify the feasibility of transitioning from microscopic structural models to macroscopic electromagnetic equivalent circuit models. More importantly, we extend classical electromagnetic equivalent circuit theory (traditionally applied to metals) to advanced carbon materials with microstructures, such as graphene films. This shift in theoretical approach not only elucidates the shielding and polarization manipulation mechanism, but also develops electromagnetic interference shielding effectiveness manipulation serialized metamaterials with some high performances: the widest polarization difference manipulation range (enhanced polarization sensitivity, 1061.60 dB/mm), the highest full polarization shielding (over 99.15%) and the most significant switching efficiency for "On" and "Off" states (3.17%-99.79%), simultaneously. This work offers a promising avenue for advancing intelligent electromagnetic shielding materials.

The thriving of communication technology simultaneously generates grave electromagnetic radiation pollution, which poses a threat to electronic instruments and human health[1-4]. Therefore, electromagnetic interference (EMI) shielding is particularly important. Nevertheless, traditional EMI shielding materials with fixed shielding effectiveness (SE), are increasingly challenged to meet the demands of sophisticated and ever-changing service scenarios[5-8], urging the development of intelligent EMI shielding materials[9-11].

So far, several strategies have been devoted to advance the prospects of intelligent EMI shielding materials, such as electrochemical potential change[1], humidity change[12], temperature change[13], and shape change[5-9,14-20], and achieved good results. However, the main obstacle to the broader utilization of the aerogel materials in the integrated electronics industry, as seen in most of the aforementioned strategies, lies in their considerable thickness of several millimeters. Furthermore,

[1]Hubei Engineering Research Center of Radio Frequency Microwave Technology and Application, State Key Laboratory of Advanced Technology for Materials Synthesis and Processing, Wuhan University of Technology, Wuhan, P. R. China. [2]School of Information Engineering, Wuhan University of Technology, Wuhan, P. R. China. ✉e-mail: zuhr@whut.edu.cn; lishuxin@whut.edu.cn; hedaping@whut.edu.cn

these strategies are susceptible to inducing irreversible degradation of EMI SE during long-term cycle manipulation process.

Recently, rotational manipulation has emerged as a widely recognized strategy capable of achieving non-attenuation EMI SE manipulation processes at a small thickness[7,10]. Wei et al. realized the rotational manipulation of EMI SE by compounding highly conductive MXene with anisotropic Wood, and the adjustable shielding efficiency was 99.8180–99.9998%[11]. Carbon fiber based materials, owing to their anisotropic structural characteristics, can attain a ΔEMI SE of 11 dB (ranging from 20 dB to 31 dB) in response to a rotation angle change from 0° to 90°[21]. However, even the minimum EMI SE still reached 20 dB, indicating a blockage of more than 99.0000% of the electromagnetic (EM) wave although in the "On" state. The anisotropic materials based on carbon nanotube (CNT) also face similar problems, thus posing challenges for achieving efficient 'On and Off' manipulation[22]. Deng et al. used screen printing technology to prepare MXene ink into films with anisotropic stripe patterns. By controlling the spacing of stripes, the lower limit of EMI shielding efficiency was reduced to 36.9043%[15]. It is proved that the lower limit of EMI shielding efficiency can be reduced by pattern design. However, the practical application to achieve efficient manipulation of EMI SE requires a further reduction of the lower limit of EMI shielding efficiency, preferably close to 0. Additionally, there remains a need for further expansion of the ΔEMI SE[23–25].

In this work, we designed a highly conductive graphene film (GAF, namely graphene assembled film) metamaterial (GAFM) and achieved large-range polarization manipulation of EMI SE allowing the switching between 'On' and 'Off' states simply by rotating GAFM. Our work not only reveals the EMI shielding mechanism and polarization manipulation mechanism of advanced carbon materials, and obtains the widest polarization difference manipulation range, the highest full polarization shielding (the EMI shielding efficiency being higher than 90% in both the X and Y polarization) and the most significant switching efficiency for "On" and "Off" states, simultaneously. Most importantly, we verify the feasibility of transitioning from the microscopic structural model of carbon materials to the macroscopic electromagnetic equivalent circuit model, which enables the manipulative design of polarization shielding of carbon-based materials. This work provides a strategy for the dynamic manipulation of electromagnetic (EM) response, and draws a blueprint for the advancement of intelligent EMI shielding materials, addressing the demands of intricate and high-fidelity electromagnetic guard systems, autonomous camouflage configurations, and future smart living environments.

## Results

### Design of GAFM for polarization manipulation of EMI SE

As shown in Fig. 1, we achieved rotational manipulation of electromagnetic interference shielding effectiveness (EMI SE) based on graphene assembled film metamaterials (GAFM). Wherein, anisotropy of GAFM is obtained by carving regular array of rectangular slits on the isotropic graphene film (GAF, namely graphene assembled film), endowing it with polarization sensitive properties. Specifically, the EMI SE is lowest when the incident electric field is oriented parallel to the short side of the slit, while the EMI SE is highest when the incident electric field polarization is orthogonal to the short side of the slit. Therefore, when the angle between the slit orientation (the long side direction) in GAFM and the direction of the electric field is rotated from 0° to 90°, the EMI SE changes from maximum value (almost completely shielding) to minimum value (almost entirely transmitting).

In order to further boost the manipulation of EMI SE, including larger polarization difference and fully polarized shielding, we compounded GAFM and CF to take advantages of these two artificial materials. When the slit orientation in GAFM and the fiber axis direction of CF is parallel (GAFM//CF), the rotational manipulation range of

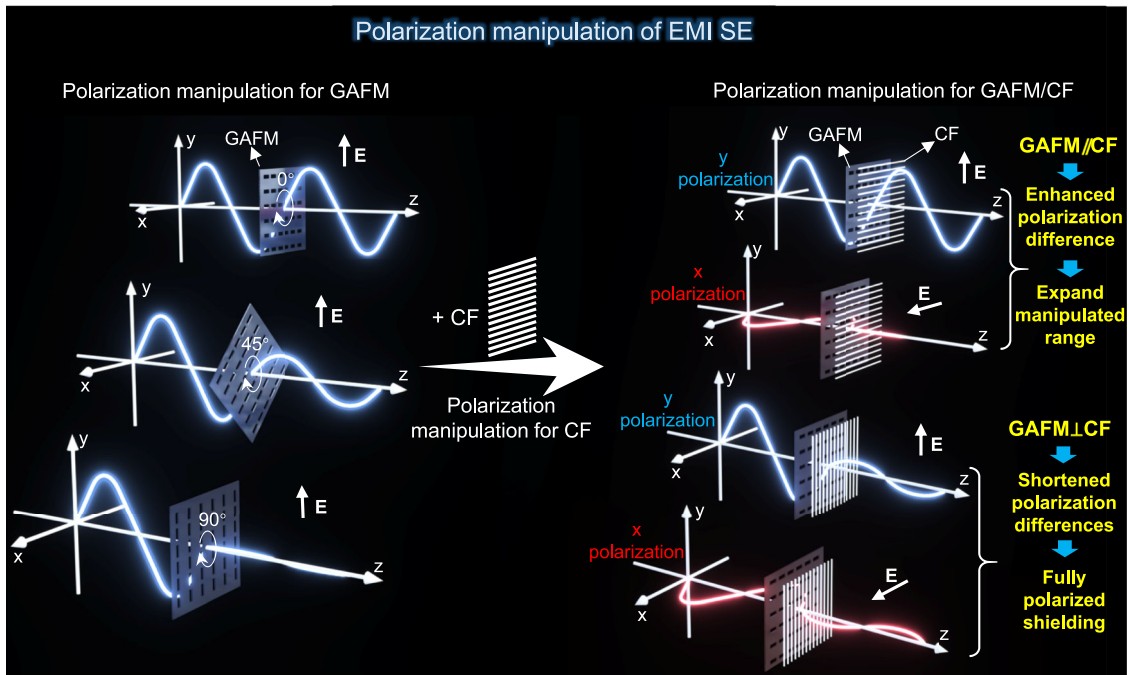

**Fig. 1 | Design of graphene assembled film metamaterials (GAFM) for polarization manipulation of electromagnetic interference shielding effectiveness (EMI SE).** Rotational manipulation of EMI SE was achieved using GAFM. Anisotropy in GAFM is induced by carving regular rectangular slits on an isotropic graphene film (GAF), imparting polarization-sensitive properties. The EMI SE is lowest when the electric field is parallel to the slit's short side and highest when orthogonal. Rotating the slit orientation from 0° to 90° changes the EMI SE from maximum (shielding) to minimum (transmission). To further enhance EMI SE manipulation, GAFM was compounded with CF. When the slit orientation in GAFM aligns with the CF fiber axis (GAFM//CF), the manipulation range is maximized, enhancing polarization differences and expanding manipulated range. When the slit orientation is perpendicular to the CF fiber axis (GAFM⊥CF), it leads to shortened polarization differences and fully polarized shielding.

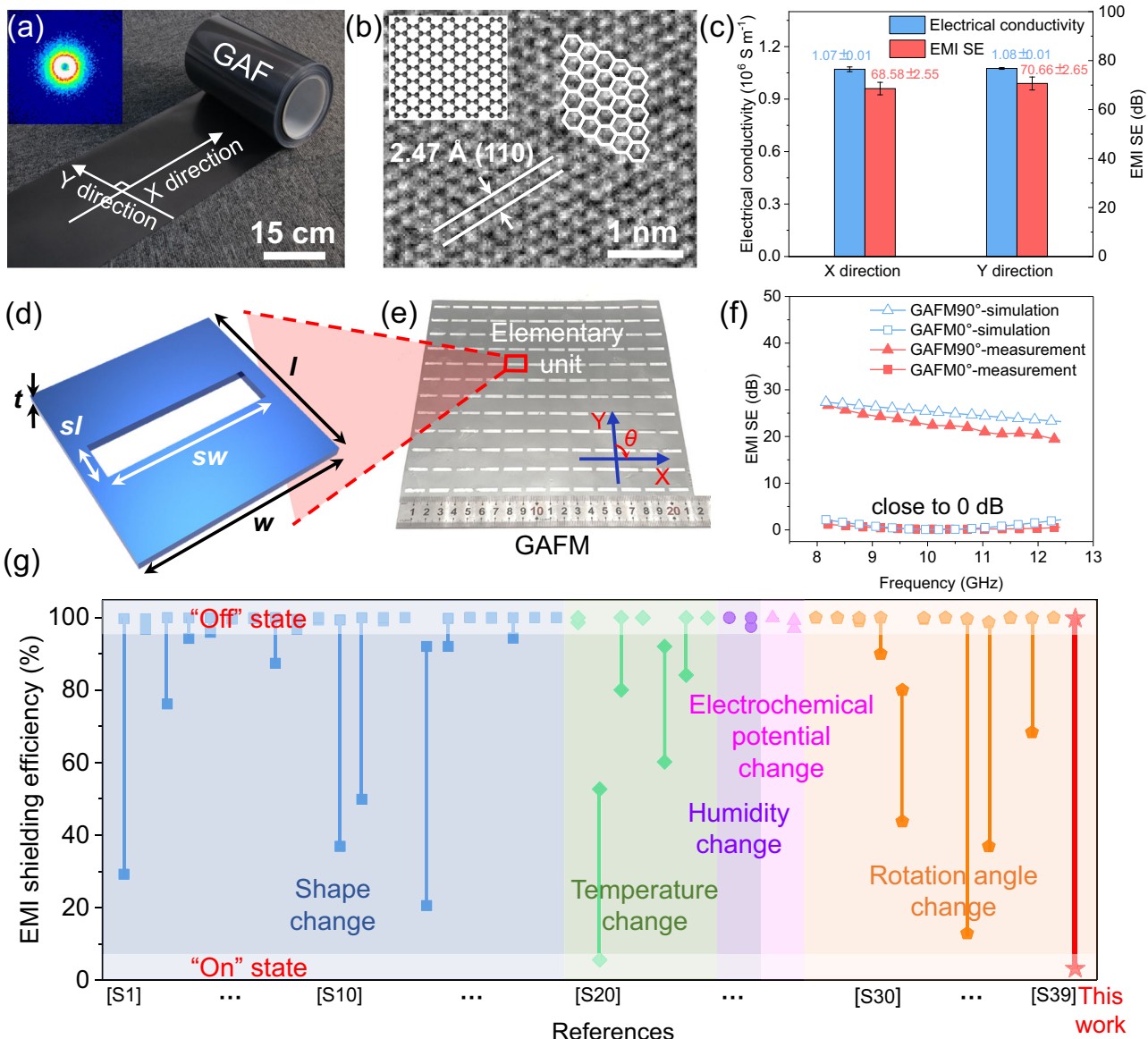

**Fig. 2 | Design and electromagnetic characteristics of anisotropic GAFM.**
**a** Digital image and small-angle X-ray scattering (SAXS) image (inset) of GAF.
**b** High-resolution transmission electron microscope (HRTEM) image and schematic diagram of the structure of graphene (inset). **c** electrical conductivity ($n = 6$) and EMI SE ($n = 10$) of GAF in $X$ and $Y$ direction. **d** Geometric parameters of the designed GAFM unit cell, with $l = 18$ mm, $w = 18$ mm, $sl = 3.5$ mm, $sw = 15$ mm, $t = 0.025$ mm. **e** Digital image of GAFM. **f** Simulated and measured EMI SE of GAFM.

**g** Comparison of EMI shielding efficiency for "On" state and "Off" state, and switching efficiency using different strategies (shape change, temperature change, humidity change, electrochemical potential change, and rotation angle change) reported in the references from [S1] to [S39] (Supplementary Information) at X band (8.2–12.4 GHz). The data are presented as mean ± s.d. The error bars represent the standard deviation from six (electrical conductivity) or ten (EMI SE) independent measurements.

GAFM/CF is maximized, achieving an enhanced polarization difference and expanding manipulated range. On the other hand, when the slit orientation in GAFM and the fiber axis direction of CF is perpendicular (GAFM⊥CF), the GAFM/CF achieves shortened polarization differences and fully polarized EMI shielding. This is also a characteristic that natural materials do not have. Hence, we believe GAFM/CF is a type of metamaterial.

**In-plane anisotropy design and polarization manipulation of GAFM**

The prepared GAF with a metallic luster possesses a highly crystalline structure (Fig. 2a), which is fabricated by high-temperature annealing followed by rolling compression treatment on graphene oxide (GO) assembly films (Supplementary Figs. 1–3). A distinct and strong (002) peak in X-ray diffraction (XRD) pattern[26–29], coupled with an ultralow

$I_D/I_G$ value of 0.057 in Raman spectroscopy (Supplementary Fig. 4)[30,31], unequivocally demonstrate high graphitization metrics of the GAF[32]. The highly graphitized structure of GAF is further confirmed by a clear crystal lattice ($d_{110} = 2.47$ Å) in Fig. 2b, which contributes to its impressive electrical conductivity value of $1.07 \times 10^6$ S m$^{-1}$ and remarkable electromagnetic interference shielding effectiveness (EMI SE) ranging from 62.48 dB to 78.23 dB in the X-band frequency range of 8.2 to 12.4 GHz (Fig. 2c, experimental details see Supplementary Note 1). Moreover, the small-angle X-ray scattering (SAXS) image of GAF reveals a symmetric circular pattern without any discernible equatorial streak scattering (inset of Fig. 2a), while the high-resolution transmission electron microscope (HRTEM) image in Fig. 2b displays an isotropic honeycomb-like structure, providing additional evidence supporting the isotropy of the GAF in the plane. Notably, the consistent performance of both the electrical conductivity and EMI SE of

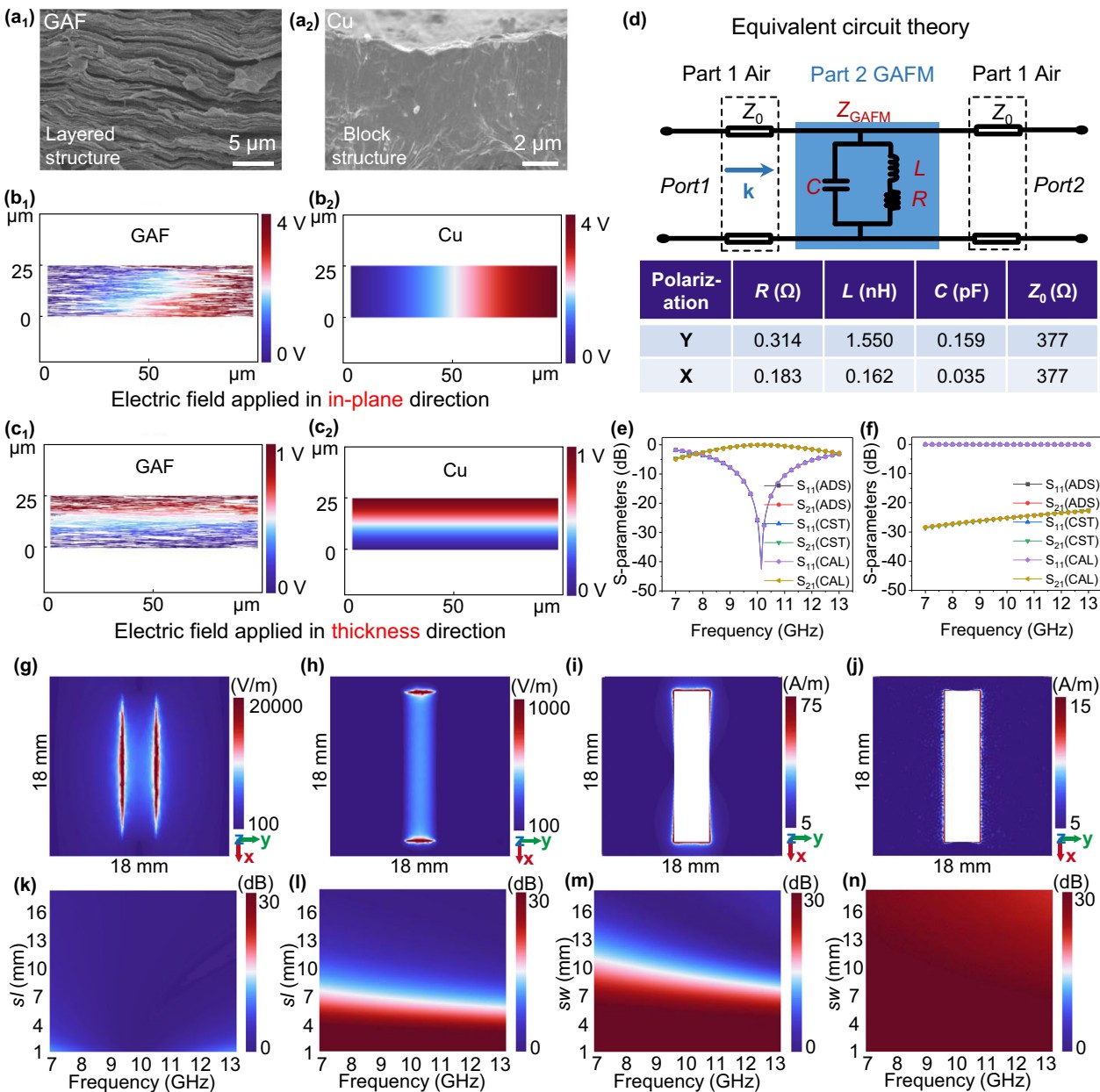

**Fig. 3 | Verification of parametric quantization and feasibility of transitioning from the microscopic structural model to the macroscopic electromagnetic equivalent circuit model.** The scanning electron microscope (SEM) cross-sectional image of ($a_1$) GAF and ($a_2$) Cu foil. Electric potential distribution of GAF when applying an electric field in the ($b_1$) in-plane direction and ($c_1$) thickness direction; electric potential distribution of Cu foil when applying an electric field in the ($b_2$) in-plane direction and ($c_2$) thickness direction. **d** Schematic diagram of equivalent circuit model and related parameters of the GAFM. Simulated transmission coefficient of the mathematical model (calculation value, named "CAL"), equivalent circuit model (Advanced Design System, named "ADS") and full wave model (Computer Simulation Technology, named "CST") for (**e**) *Y* polarization direction and (**f**) *X* polarization direction. Field map of electric field distribution of GAFM for (**g**) *Y*-polarized and (**h**) *X*-polarized electromagnetic waves at 10 GHz. Field map of surface current distribution of GAFM for (**i**) *Y*-polarized and (**j**) *X*-polarized electromagnetic waves at 10 GHz. Dependence of ΔEMI SE in (**k**) *Y* polarization and (**l**) *X* polarizations on the lengths of *sl* at 7–13 GHz (*sw* = 15.0 mm). Dependence of ΔEMI SE in (**m**) *Y* polarization and (**n**) *X* polarizations on the lengths of *sw* at 7–13 GHz (*sl* = 3.5 mm).

the GAF in both the X and Y directions further confirms its isotropic nature within the plane (Fig. 2c, Supplementary Fig. S5).

Through geometric design of unit cell as shown in Fig. 2d, the isotropic GAF is transformed into the anisotropic GAF-based meta-material in the plane for electromagnetic properties. The GAFM with overall size of 216 mm × 216 mm (Fig. 2e) was obtained by laser engraving on GAF according to the designed geometric parameters (Supplementary Fig. 6). Figure 2f exhibits the measured and simulated EMI SE of GAFM at X band before and after rotating 90°. When the rotating angle is 0° (defined as when the incident electric field is oriented parallel to the short side of the slit), the EMI SE of GAFM

diminishes to a minimum of 0.14 dB (shielding efficiency of 3.17%), closely approximating 0 dB and reaching the lowest value reported in references (Details in Supplementary Table 1), thus facilitating efficient 'On and Off' manipulation (Supplementary Fig. 7). Conversely, when the rotating angle is 90° (defined as when the incident electric field is oriented parallel to the long side of the slit), the EMI SE of GAFM consistently maintains above 20 dB over the entire frequency band, peaking at 26.68 dB, thereby indicating a maximum shielding efficiency of 99.79%. This occurs because when the length of the periodic metamaterial structure equals half of the corresponding wavelength, the wave antinodes align spatially, causing both the voltage and

current amplitudes to reach their maximum values, which signifies that the system has entered a resonant state (passband). For different polarizations, the side lengths in the X and Y directions are different, which results in different resonant frequencies, thereby obtaining a designable high polarization sensitivity at the same operating frequency (See Supplementary Note 2).

The GAFM achieved a minimum and a maximum electromagnetic interference (EMI) shielding effectiveness (SE) of 0.14 dB and 26.68 dB (experimental details see Supplementary Note 3), corresponding to high electromagnetic wave transmittance of 96.83% and high EMI shielding efficiency of 99.79%, respectively, showing the most significant polarization switching effect. As shown in Fig. 2g, these two properties of GAFM show obvious advantages compared with the values reported in existing references for rotation angle change strategy and other strategies.

## Extension of classical electromagnetic equivalent circuit theory to GAFM with microstructures

In addition, the excellent performance of GAFM is mainly attributed to the ingenuity of design parameters. Firstly, we confirmed the layered microstructure of GAF and the block structure of Cu foil based on the cross-sectional image of scanning electron microscope (SEM) in Fig. 3a, and modeled and simulated their microstructures. According to the simulation results (Fig. 3b, c), we found that the electric field distribution of GAF in the plane and thickness direction is both consistent with these of Cu foil, which meets the application premise of the classical equivalent circuit theory of metal. Then, we modeled the GAFM as a combination of capacitance-inductance-resistance and calculated parameters of GAFM based on the equivalent circuit model (Fig. 3d). It is worth noting that the GAFM is a pure conductive film without any substrate. Therefore, in this work, the equivalent circuit model is established using air as the substrate for the corresponding transmission line[33]. According to transmission line theory, the transfer matrix of this structure can be expressed as

$$
\begin{bmatrix} A & B \\ C & D \end{bmatrix} = \begin{bmatrix} \cos\theta & jZ_0\sin\theta \\ j\frac{1}{Z_0}\sin\theta & \cos\theta \end{bmatrix} \cdot \begin{bmatrix} 1 & 0 \\ j\omega C & 1 \end{bmatrix} \cdot \begin{bmatrix} 1 & 0 \\ \frac{1}{R+j\omega L} & 1 \end{bmatrix}
$$
$$
\cdot \begin{bmatrix} \cos\theta & jZ_0\sin\theta \\ j\frac{1}{Z_0}\sin\theta & \cos\theta \end{bmatrix} = \begin{bmatrix} -1 & -jZ_0^2\omega C - \frac{Z_0^2}{R+j\omega L} \\ 0 & -1 \end{bmatrix} \quad (1)
$$

where $\theta = \pi/2$, $Z_0 = 377\,\Omega$.

Its scattering matrix can be expressed as

$$
\begin{bmatrix} S_{11} & S_{12} \\ S_{21} & S_{22} \end{bmatrix} = \begin{bmatrix} \frac{A+B/Z_0-CZ_0-D}{A+B/Z_0+CZ_0+D} & \frac{2(AD-BC)}{A+B/Z_0+CZ_0+D} \\ \frac{2}{A+B/Z_0+CZ_0+D} & \frac{-A+B/Z_0-CZ_0+D}{A+B/Z_0+CZ_0+D} \end{bmatrix} \quad (2)
$$

Its impedance matrix can be expressed as

$$
\begin{bmatrix} Z_{11} & Z_{12} \\ Z_{21} & Z_{22} \end{bmatrix} = \begin{bmatrix} \frac{A}{C} & \frac{AD-BC}{C} \\ \frac{1}{C} & \frac{D}{C} \end{bmatrix} \quad (3)
$$

The input impedance can be expressed as

$$
Z_{in} = \frac{AZ_0+B}{CZ_0+D} \quad (4)
$$

To validate the precision of the proposed equivalent circuit model (ECM) concerning the design parameters of the GAFM, we respectively calculated the response of the equivalent circuit using three different methods: the two-port microwave network mathematical calculation (referred to as "CAL"), the circuit simulation software Advanced Design System (ADS), and the full-wave electromagnetic simulation software CST Microwave Studio. Among them, the parameters of circuit

component for the Y polarization direction are as follows: $R = 0.314\,\Omega$, $L = 1.550$ nH, $C = 0.159$ pF, $Z_0 = 377\,\Omega$. For the X polarization direction, the parameters of circuit component are as follows: $R = 0.183\,\Omega$, $L = 0.162$ nH, $C = 0.035$ pF, $Z_0 = 377\,\Omega$. Based on the results of Figs. 3e, f, we found that the simulated transmission coefficient of the mathematical model, equivalent circuit model and full wave model are consistent for both Y polarization direction and X polarization direction, verifying that the design of GAFM is feasible from theory to practice. Finally, the field map verifies the series of results and proves that the design principle of the dimension of targeted sills is effective, as illustrated in Fig. 3g–j. We verified the feasibility of transitioning from a microscopic structural model of carbon materials to the macroscopic electromagnetic equivalent circuit model. This shift not only facilitates the manipulation of polarization shielding, but also opens the door to a deeper understanding of the material properties at the macroscopic scale, showing the great potential of extending from metal to carbon material to become the next generation electromagnetic wave response manipulation material.

Based on the above theoretical guidance, the manipulation and design of the EMI SE of GAFM under specific polarization can be realized through changing the structural size parameters. Figure 3k, l show the dependence of the EMI SE in Y polarization and X polarization on the length of sl at 7–13 GHz when the slit width is fixed (sw = 15.0 mm), while Fig. 3m, n show the dependence of the EMI SE in Y polarization and X polarization on the length of sw at 7–13 GHz when the slit length is fixed (sl = 3.5 mm). The EMI SE of GAFM in X-band at Y polarization and X polarizations varies with the change of polarization angle (Supplementary Fig. 8). Therefore, the EMI SE is almost only manipulated by the size parameters parallel to the polarization direction, which proves the anisotropy of the electromagnetic shielding of this structure. Moreover, we attempted to obtain GAFM-T by typically setting geometric parameters. The geometric parameters of GAFM-T and the EMI SE at different rotation angles are shown in Supplementary Fig. 9. Obviously, the ability of GAFM-T to manipulate EMI SE, such as the minimum EMI SE and the manipulation range of ΔEMI SE, is inferior to those of the proposed GAFM. To effectively analyze the non-specular reflection modes of the GAFM (Supplementary Fig. 10), we conducted simulations to examine the far-field radiation patterns of a 10 × 10 GAFM array. Plane electromagnetic waves polarized in the Y and X directions were used as excitation sources.

Under conditions of normal incidence, the primary reflected beam of the GAFM array aligns perpendicularly with the incident direction, thereby demonstrating the absence of non-specular reflection phenomena. Additionally, at 10 GHz, the intensity of the reflected beam for Y-polarized waves is significantly lower than at 8 GHz and 12 GHz, while X-polarized waves consistently exhibit higher amplitude reflections. These results align with individual element simulations. For further analysis, a horn antenna was used as the feed source in simulations. Supplementary Fig. 11a–c show the far-field radiation patterns of the horn antenna, and Supplementary Fig. 11d–i depict the far-field beams of the GAFM array with the horn. The results obtained from horn feeds with different polarization orientations are consistent with those derived from plane wave simulation, confirming the reliability of our approach.

## Further polarization manipulation of GAFM/CF

To further expand the EMI SE polarization manipulation difference (ΔEMI SE), we composited carbon fiber (CF) material with GAFM. As illustrated in Fig. 4a, CF exhibits a highly oriented structure. We define the direction of CF as 0° when the CF orientation is perpendicular to the incident electric field polarization. With the gradual increase in rotation angle, the EMI SE of CF gradually improves, as depicted in Fig. 4b.

To achieve expanded polarization difference range and fully polarized shielding, we employed two corresponding modes: parallel

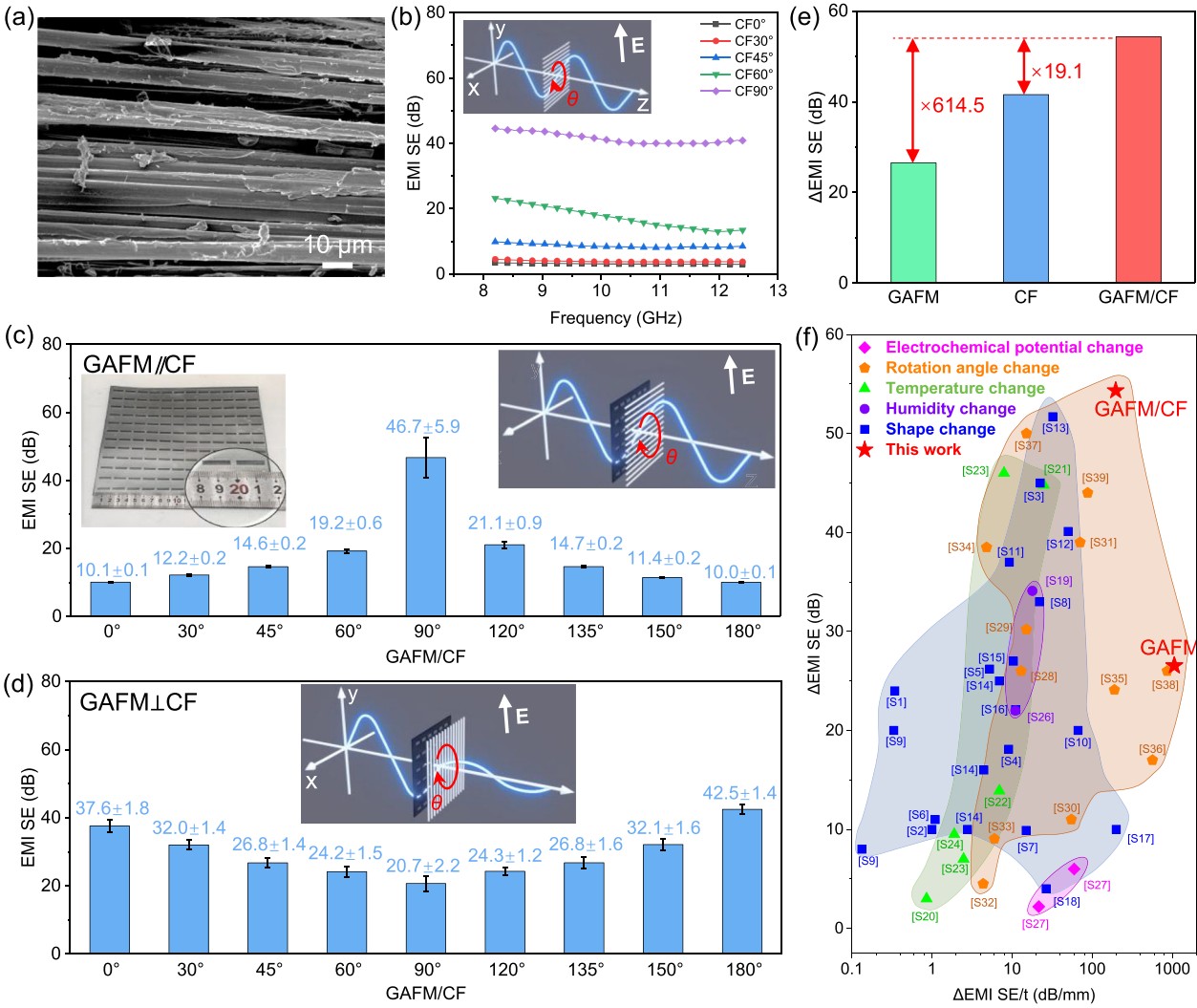

**Fig. 4 | Polarization manipulation of EMI SE with the composite structure of GAFM and carbon fiber (GAFM/CF). a** SEM image and (**b**) EMI SE of CF. EMI SE manipulation of GAFM/CF in (**c**) parallel ($n = 10$) and (**d**) orthogonal states ($n = 10$). **e** ΔEMI SE of GAFM/CF. **f** Comparison of the EMI SE polarization manipulation difference (ΔEMI SE) and ΔEMI SE per unit thickness (ΔEMI SE/t) using different strategies (electrochemical potential change, rotation angle change, temperature change, humidity change and shape change) reported in the references from [S1] to [S39] (Supplementary Information). The data are presented as mean ± s.d. The error bars represent the standard deviation from ten independent measurements.

combination (GAFM//CF, Supplementary Fig. 12) and vertical combination (GAFM⊥CF, Supplementary Fig. 13), respectively. Upon increasing the rotation angle of GAFM//CF from 0° to 90°, and subsequently to 180°, the EMI SE within the X band shows a trend of initial increase followed by a decrease (Fig. 4c). The polarization shielding difference is notably enhanced to 54.33 dB, which represents a significant enhancement in polarization difference. Conversely, when the rotation angle of GAFM⊥CF increases from 0° to 90°, and further to 180°, the EMI SE within the X band demonstrates a trend of initial decrease followed by an increase (Fig. 4d). Most importantly, the GAFM⊥CF case achieves a smaller EMI SE difference with the full-angle EMI SE exceeding 20.7 dB (full-angle EMI shielding over 99.15%), indicating fully polarized EMI shielding. Furthermore, we investigated the variation of EMI SE with increasing rotation angle in the GAFM-rotated/CF-fixed state. As shown in Supplementary Fig. 14, the EMI SE difference also remains small levels.

Overall, the composite structure of GAFM and CF enables enhanced polarization differences and full polarization shielding. Additionally, the EMI SE range of GAFM/CF spans from 9.90 dB to 64.23 dB (Supplementary Fig. 15), significantly surpassing that of GAFM (ranging from 0.14 dB to 26.68 dB) and CF (ranging from 2.97 dB to 44.54 dB). Consequently, the ΔEMI SE of GAFM/CF attains 614.5 times that of GAFM and 19.1 times that of CF, respectively (Fig. 4e). Notably, when compared to various intelligent EMI shielding materials reported in the references (Details in Supplementary Table S2), the ΔEMI SE per unit thickness (ΔEMI SE/t) of our designed and prepared GAFM reaches a high value of 1061.60 dB/mm, while the ΔEMI SE of GAFM/CF attains a high value of 54.33 dB (Fig. 4f, Supplementary Table 2).

**Polarization manipulation mechanism of GAFM and GAFM/CF**

The maneuverable EMI SE of GAFM and CF stems from their anisotropic structures. In accordance with Maxwell's equations, incident space EM waves induce surface current density on the surface of a conductor (Fig. 5a–c)[15]. If the current is cut off, the electric field and space EM waves will be regenerated (transmission). If the electromagnetic is not cut off, the electromagnetic energy exists in the form of current without generating an electric field (reflection). In the case of, alignment of direction of GAFM parallel to the incident EM wave's electric field maximizes the continuities of conductive path, enhancing surface current density generated on GAFM's surface and thus improving EM wave attenuation efficiency (Fig. 5a, b). Conversely,

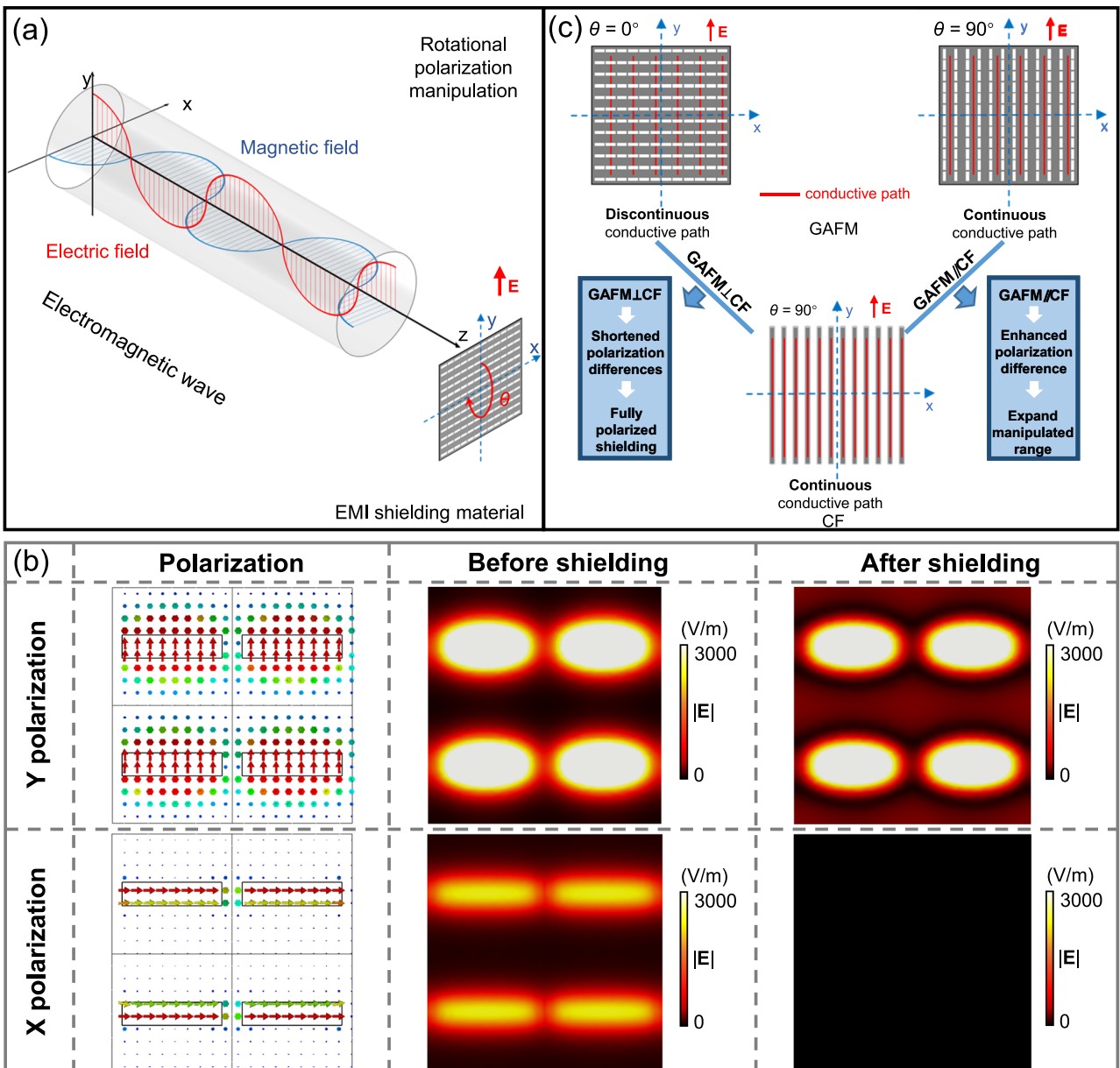

**Fig. 5 | Schematic diagram of the EMI SE manipulation mechanism of the composite structure of GAFM and CF (GAFM/CF) in parallel and orthogonal states. a** Schematic diagram illustrating the electric and magnetic fields of incident electromagnetic (EM) waves and the angle between the direction of shielding material and the electric field's direction. **b** Vector electric field of incident electromagnetic wave and simulated electric field distributions before and after GAFM shielding in *X* and *Y* polarizations of electric field, respectively. **c** Distribution of conductive paths for GAFM at 0° and 90° rotation angle, as well as the distribution of conductive paths for GAFM and CF in perpendicular and parallel states, respectively.

when GAFM is perpendicular to the electric field direction, discontinuities in the conductive path substantially reduce induced current density, resulting in weak attenuation of EM waves. The CF with obvious anisotropic structure has EMI SE rotation manipulation effect similar to GAFM (Supplementary Fig. 16).

Figure 5b shows a vector electric field distribution of incident electromagnetic wave and compares the EMI SE described by electric field amplitudes (|**E**|) under *X* polarization (defined as when the incident electric field is oriented parallel to the long side of the slit) and *Y* polarization (defined as when the incident electric field is oriented parallel to the short side of the slit), respectively. For *Y* polarization of the electric field, |**E**| remains virtually unchanged both before and after passing through GAFM, which means the undiminished electric field strength and electromagnetic energy. Conversely, for X polarization of the electric field, |**E**| transitions

from a strong state to a weak disappearing state before and after traversing the GAFM, indicating that most of the electromagnetic energy is efficiently shielded. Consequently, the simulation outcomes presented in Fig. 5b align consistently with the data depicted in Fig. 2f.

To enhance EMI SE manipulation effects, such as increasing polarization difference and achieving fully polarized shielding, we compounded GAFM and CF to leverage their respective strengths (Fig. 5c). When the slits in GAFM and the fiber axis direction of CF are aligned parallelly (GAFM//CF), it maximizes the rotational manipulation range of GAFM/CF, thereby enhancing polarization difference and extending the manipulation range. Conversely, when the slits in GAFM and the fiber axis direction of CF are perpendicular (GAFM⊥CF), GAFM/CF achieves reduced polarization differences while achieving fully polarized EMI shielding.

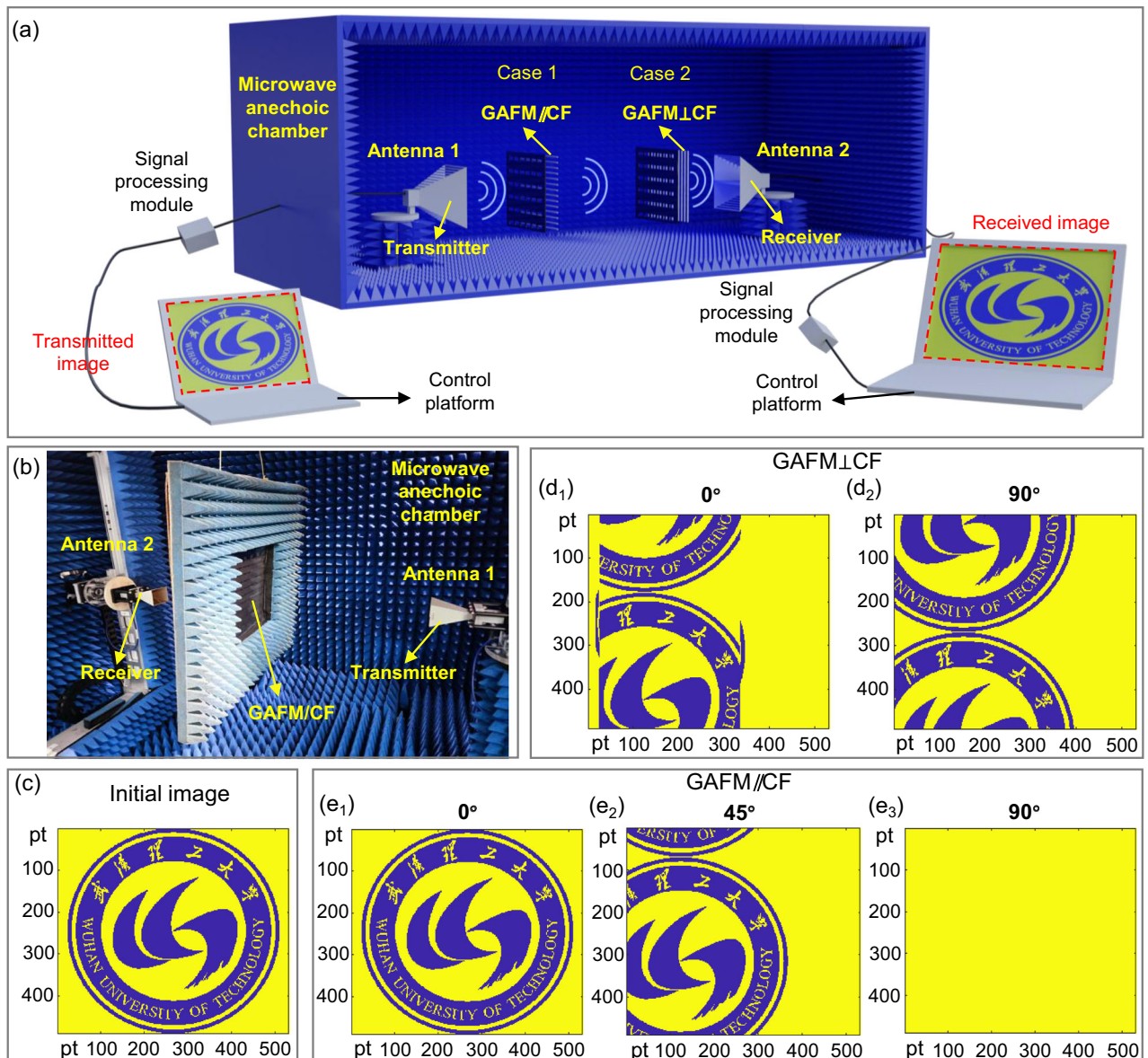

**Fig. 6 | Application demonstration of polarization manipulation of EMI SE with the composite structure of GAFM and carbon fiber (GAFM/CF) in information transmission.** Application demonstration of polarization manipulation of EMI SE with GAFM/CF in information transmission. **a** Schematic of information communication scheme based on GAFM/CF, in which a color image is transmitted from the transmitter to the receiver with a GAFM/CF barrier under different composite cases and distinct rotation angles. **b** Actual scene diagram of measuring instrument. **c** Origin state of a color image before being transmitted. **d₁, d₂** Color images received by GAFM⊥CF case at different rotation angles (0° and 90°). **e₁–e₃** Color images received by GAFM//CF case at different rotation angles (0°, 45° and 90°).

## Application display of GAFM/CF

To validate the effects of rotation manipulation of EMI SE, we further applied GAFM/CF into the wireless information transmission scene (Fig. 6a). As shown in Fig. 6b, data transmission was carried out by a practical wireless communication system. In this scene, the manipulation platform digitized the color image (Fig. 6c, the logotype of Wuhan University of Technology), then the signal processing module transforms the data into the radiofrequency (RF) signal, which is thereafter fed into the receiver. For the barrier, we took 2 cases (GAFM//CF and GAFM⊥CF) and obtained different effects. The GAFM/CF prototype of different cases was measured in a microwave anechoic chamber using an electromagnetic near-field measurement system with a vector network analyzer (VNA, Agilent Technologies N5247A). The measurement configuration involved a pair of linearly polarized horn antennas with identical specifications aligned at their phase center to ensure consistency and a middle-placed GAFM/CF prototype. This setup was implemented to investigate the prototype's performance. During the measurement, the electromagnetic polarization sensitivity characteristics of the prototype under measurement were assessed by synchronously rotating the two receiving and dispatching horn antennas. This methodology allowed for accurate measurement of the prototype's capabilities which enabled meaningful conclusions.

As illustrated in Fig. 6d₁, d₂, two images (GAFM⊥CF, rotate overall by 0° and 90° respectively) both have a certain degree of distortion, indicating that GAFM⊥CF case achieves full polarization shielding, greatly weakening the signal strength. While in GAFM//CF case, Fig. 6e₁–6e₃ (GAFM//CF, rotate overall by 0°, 45° and 90° respectively) exhibited a significant difference. Only when GAFM//CF rotate 0°, the image quality is well maintained, and as the rotation angle increases, the distortion becomes more severe. Thus, GAFM//CF case achieved the widest polarization manipulation range. These results are consistent with the conceptual framework of Fig. 1, and show great

potential in complex and high assurance EM guard devices, tunable setups, and future smart life[34–36].

## Discussion

In summary, we developed a highly conductive GAFM capable of achieving polarization manipulation of EMI SE through rotating. The fabrication of GAFM involved simulation-driven design coupled with precise laser engraving techniques, realizing a minimum EMI SE of 0.14 dB and a high ΔEMI SE/t value of 1061.60 dB/mm. By strategically integrating GAFM with CF and meticulous manipulating the orientation of both materials in parallel and vertical configurations, the GAFM/CF composite based on an all-carbon material reached an historic pinnacle in the manipulation range of EMI SE (54.33 dB), and fully polarized shielding (full-angle EMI shielding over 99.15%), respectively. This work offers a significant conceptual advance in the demonstration of the feasibility of transitioning from the microscopic structural model of carbon materials to the macroscopic electromagnetic equivalent circuit model, thus obtaining the excellent shielding performance of carbon-based materials, which enables the manipulative design of polarization shielding of carbon based materials, an achievement that has not been previously demonstrated. Previous research on EMI SE manipulation in carbon materials has been limited by the lack of a solid electromagnetic theory and model. Our work bridges this gap and offers a path toward more effective and high-performance EMI SE manipulation. This work establishes a paradigm for dynamic manipulation of EMI SE through rotation strategy, delineating an efficacious pathway towards realizing high-performance dynamic manipulation over electromagnetic wave responses and advanced fields of intelligent EMI shielding and associated domains.

## Methods
### Chemicals
The graphene oxide (GO) filter cake (50 wt.%) was purchased from the Wuxi Chengyi education technology Co., Ltd. Supplementary Fig. 2 illustrates detailed characterizations of the GO nanosheets.

### Material Syntheses
**Preparation of GAF**. The GAF was prepared by thermal reduction of GO. Firstly, GO filter cake was dispersed with ultrapure water (3 wt.%) by a homogeneous mixer. Secondly, GO assembled film was fabricated by coating on a substrate followed by drying process. Thirdly, GO assembled film was peeled off from the substrate and subjected to high-temperature heat treatment (3000 °C for 1 h). Finally, the GAF was obtained by applying a pressure of 300 MPa. Detailed characterizations of the GAF are presented in Supplementary Fig. 3 and Supplementary Fig. 4.

**Design of the GAFM**. The simulation of the GAFM is mainly carried out using CST Microwave Studio software. The design of the metasurface follows the design principle of selecting the surface with bandpass frequency. The geometric parameters of the designed unit cell in GAFM are $l = 18$ mm, $w = 18$ mm, $sl = 15$ mm, $sw = 3.5$ mm and $t = 0.025$ mm (Figs. 2d, 3a). As a comparison, we added a set of comparisons (GAFM-T), and the geometric parameters of unit cell in GAFM-T are $l = 5$ mm, $w = 5$ mm, $sl = 1$ mm, $sw = 4$ mm and $t = 0.025$ mm (Supplementary Fig. 9).

**Preparation of the GAFM**. The geometric dimensions of the pre-designed GAFM were fed into the LPKF CircuitPro PL 2.0 computational simulation software. Subsequently, using a laser engraving machine (LPKF Laser & Electronics ProtoLaser S), GAFM was manufactured in one step according to the calculated laser path after removing the graphene material from the designed slits (Supplementary Fig. 6).

**Preparation of the CF**. The CF film was prepared from polyacrylonitrile (PAN) as precursor. PAN fiber was pre-oxidized in air at 300 °C. Then the pre-oxidized fiber was carbonized in inert argon atmosphere at high temperature of 1200 °C, and then introduced into release paper coated with epoxy resin for rolling, and finally the CF film was obtained. Among them, a single carbon fiber has a diameter of about 7 μm and a density of 1.8 g cm$^{-3}$. The electrical conductivity of the CF films is $589 \pm 95$ S m$^{-1}$.

**Preparation of the GAFM/CF**. The GAFM/CF composite films were prepared by bonding GAFM and CF films together with adhesive.

### Simulation
**Microscopic structural simulation**. Based on the layer stacking structure of GAF observed in the cross-sectional SEM image and ultrahigh conductivity, a microscopic model was established. Among them, the layered structure simulates the microstructure of GAF, while the rectangular blocks represent the Cu metal blocks as the control group. The electrical boundary conditions are as follows: (1) for the electric field applied in the thickness direction, the lower boundary is grounded with a potential of 0 V, and the upper boundary potential is assigned a potential of 1 V; (2) for the electric field applied in the in-plane direction, the left boundary is grounded with a potential of 0 V, and the right boundary has a potential of 4 V. The conductivity of GAF is calculated to be $1.068 \times 10^6$ S m$^{-1}$, which is in close agreement with the actual value ($1.07 \times 10^6$ S m$^{-1}$). Solve in frequency domain and derive electric potential distribution results.

**Macroscopic electromagnetic equivalent circuit model simulation**. The design and simulation of the GAFM was performed using CST Microwave Studio. The unit cell structure is simulated with periodic boundary conditions along the $x$ and $y$ axes, while Floquet port excitations are applied along the $z$ direction, both above and below the unit cell. The GAFM was modeled as a combination of capacitance-inductance-resistance and parameters of GAFM based on equivalent circuit model was calculated. To effectively analyze the non-specular reflection modes of the GAFM, simulations was conducted by CST to examine the far-field radiation patterns of a $10 \times 10$ GAFM array. Plane electromagnetic waves polarized in the Y and X directions were used as excitation sources. Under conditions of normal incidence, the primary reflected beam of the GAFM array aligns perpendicularly with the incident direction, thereby demonstrating the absence of non-specular reflection phenomena. For further analysis, a horn antenna was used as the feed source in simulations, and the the far-field radiation patterns of the horn antenna, and the far-field beams of the GAFM array with the horn were obtained from horn feeds with different polarization orientations.

**Wireless image transmission measurement of the GAFM/CF**. Wireless image transmission was carried out by a tangible wireless communication system. In this scene, the manipulation platform digitized the color image with a 534 pt × 490 pt pixel (the logotype of Wuhan University of Technology), then the signal processing module transforms the data into the RF signal, thereafter the RF signal is sent to the receiver. For the barrier, two cases (GAFM//CF and GAFM⊥CF) were took and different effects were obtained.

### Statistics and reproducibility
Data are representative of five independent experiments with similar results.

### Reporting summary
Further information on research design is available in the Nature Portfolio Reporting Summary linked to this article.

## Data availability

The data that support the plots are available within this paper and its Supplementary Information. The Figs. 2–4, and Supplementary Figs. 2, 4, 5, 7, 9, 12, 13, 14, 15, 17, 21, 26 data generated in this study have been deposited in the zenodo database under accession code https://doi.org/10.5281/zenodo.17361341 Source data are provided with this paper.

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

## Acknowledgements

This work was supported by the National Natural Science Foundation of China (62401413 received by H.R.Z., 22279097 received by D.P.H.), Natural Science Foundation of Hubei Province (2025AFB038 received by Z.W.), Foundation of National Key Laboratory of Microwave Imaging Technology (received by Z.W.), Foshan Xianhu Laboratory Project (XHD2024-12134142-02 received by S.X.L.), and Fundamental Research Funds for the Central Universities (WUT: 2024IVA031 received by H.R.Z.).

## Author contributions

Z.W., H.R.Z. and D.P.H. designed the project. Z.W., Z.L., L.Z., J.K.X. and Z.X.Z. performed the preparation and processing of materials and data analysis. Z.L., W.Q., H.Q.F., L.L., H.F., P.F.C., L.Z. and H.Y. carried out the characterization and measurement. H.R.Z., Y.Z., Y.P.R. and H.Z.Z. performed the simulation calculation. H.R.Z., X.Z., S.X.L., and D.P.H. reviewed the manuscript. S.X.L. and D.P.H. supervised the whole process. All authors participated in the discussion of the results, commented on the implications, and fully approved the content of the manuscript.

## Competing interests

The authors declare no competing interests.
