## [Transparent Peer Review file · Nature Communications]

Polarization Manipulation of Electromagnetic Interference Shielding Effectiveness Utilizing Graphene Film-Based Metamaterials

Corresponding Author: Professor Daping He

Version 0:

Reviewer comments:

Reviewer #1

(Remarks to the Author)

The manuscript developed a GAFM capable of achieving polarization manipulation of EMI SE through rotating. The design is effective, obtaining a quite high specific EMI SE polarization difference. Below addressed some points need to be further clarified. Before publication, major revision needs to be done.

1. The Introduction is insufficient to review the present work regarding the rotational modulation of EMI SE. Focusing on what has been done on the rotational modulation of microwave will be more insightful. There are plenty of work on tailoring the microwave by using the anisotropic material.
2. In section 'Design of GAFM for polarization manipulation of EMI SE', please give the design principle of the dimension of tailored slits.
3. What type of CF film has been used in this study? What's the electrical conductivity of the CF films?
4. Please provide the unit cell parameter optimization results in the supporting information for analyzing the size effect on the EMI SE.
5. Figure 3(f) gives the Δ EMI SE and Δ EMI SE/t comparison of different strategies. Please add results of other rotational modulation results as well.
6. In Figure 4, except for the schematic diagram, please add the CST simulation diagram to strengthen the mechanism analysis.

Reviewer #2

(Remarks to the Author)

The unit cell dimension is 18 mm, which is around $\lambda/2$ in comparison with the operating band. This is a very well-known fact, there is no metamaterial.

Further, this may invoke non-specular modes which must be investigated by the authors.

What is the role of graphene in the design? What will happen if the graphene is replaced by conventional metal or dielectric?

How the graphene layer has been prepared? How they have been modelled theoretically?

The experimental frequency responses have not been shown.

How the structure is advantageous to the reported ones?

Reviewer #3

(Remarks to the Author)

The paper reports the development of thin EMI shielding film using a highly conducting graphene sheet. The EMI mechanism is altered by making slots in the conducting sheet. Additionally, an oriented CF sheet is used to reflect the EM radiation. The cross structure between graphene slots and the alignment of CF gives the polarisation rotation. The paper can not be acceptable in its present form for the following reasons. Apart from lacking novelty, the work does not quantify many parameters. For example, the relation between the width of the slot and the efficiency of polarisation rotation/shielding efficiency etc. There is no explanation why the combination is called "metamaterial".

Version 1:

Reviewer comments:

Reviewer #2

(Remarks to the Author)

Where is the role of metamaterials in the design? The periodicity of the structure is 18 mm, while the device frequency is in X-band. Thus, the dimension is not in quarter-wavelength calling for possibility of spurious modes which has not been studied.

The EMI measurements should be carried out step by step and shown accordingly.

The equivalent circuit design mentioned in Fig. 3 is not correct as the wave impinges on the structure, it undergoes transmission along the substrate; thus, there needs a consideration of an equivalent transmission line. See <https://doi.org/10.1364/AO.431821> for details.

The circuit model response and the simulated/ measured results of the S-parameters must be compared.

A comparison table highlighting the novelty of the design compared to the existing one should be mentioned.

Reviewer #3

(Remarks to the Author)

The authors have addressed the concerns and therefore, the paper can be accepted in its present form.

Version 2:

Reviewer comments:

Reviewer #2

(Remarks to the Author)

Reviewer #4

(Remarks to the Author)

In my opinion, the authors have adequately addressed referees' comments. Therefore, it can be accepted in the present form.

Author's Responses to Comments

Dear reviewers,

The revised manuscript has been greatly benefited from your constructive comments and valuable suggestions. We would like to thank you sincerely for the time and effort to help improve the presentation and qualification of this manuscript.

Response to Reviewer #1

General comment:

The manuscript developed a GAFM capable of achieving polarization manipulation of EMI SE through rotating. The design is effective, obtaining a quite high specific EMI SE polarization difference. Below addressed some points need to be further clarified. Before publication, major revision needs to be done.

REPLY:

Firstly, we would like to thank you sincerely for the time and effort spent to help improve the presentation of this paper. Heartily speaking, your constructive comments and suggestions are very professional, which help us to articulate the potential of the proposed work and highlight the strengths. We are deeply grateful for your contribution.

Original comment 1-1:

The Introduction is insufficient to review the present work regarding the rotational modulation of EMI SE. Focusing on what has been done on the rotational modulation of microwave will be more insightful. There are plenty of work on tailoring the microwave by using the anisotropic material.

REPLY:

We thank the reviewer for this valuable advice. We have revised the Introduction and supplemented more descriptions about the present research work regarding the rotational manipulation of EMI SE in the revised manuscript (Page 3-4, Line 45-70) as follows:

“So far, several strategies have been devoted to advance the prospects of intelligent EMI shielding materials, such as electrochemical potential change,¹ humidity change,¹² temperature change,¹³ and shape change,^{5-9,14-20} and achieved good results. However, the main obstacle to the broader utilization of the aerogel materials in the integrated electronics industry, as seen in most of the aforementioned strategies, lies in their considerable thickness of several millimeters. Furthermore, these strategies are susceptible to inducing irreversible degradation of EMI SE during long-term cycle manipulation process.

Recently, rotational manipulation has emerged as a widely recognized strategy capable of achieving non-attenuation EMI SE manipulation processes at a small thickness.^{7,10} Wei et al. realized the rotational manipulation of EMI SE by compounding highly conductive MXene with anisotropic Wood, and the adjustable shielding efficiency was 99.8180%~99.9998%.¹¹ Carbon fiber based materials, owing to their anisotropic structural characteristics, can attain a Δ EMI SE of 11 dB (ranging from 20 dB to 31 dB) in response to a rotation angle change from 0° to 90°.²¹ However, even the minimum EMI SE still reached 20 dB, indicating a blockage of more than 99.0000% of the electromagnetic (EM) wave although in the “on” state. The anisotropic materials based on carbon nanotube (CNT) also face similar problems, thus posing challenges for achieving efficient ‘on and off’ manipulation.²² Deng et al. used screen printing technology to prepare MXene ink into films with anisotropic stripe patterns. By manipulating the spacing of stripes, the lower limit of EMI shielding efficiency was reduced to 36.9043%.¹⁵ It is proved that the lower limit of EMI shielding efficiency can be reduced by pattern design. However, the practical application to achieve efficient manipulation of EMI SE requires a further reduction of the lower limit of EMI shielding efficiency, preferably close to 0. Additionally, there remains a need for further expansion of the Δ EMI SE.^{23-25”}

Original comment 1-2:

In section 'Design of GAFM for polarization manipulation of EMI SE', please give the design principle of the dimension of tailored slits.

REPLY:

Thank you for your valuable advice. We have supplemented the train of thought and the related details of the design in the revised manuscript (Page 9-13, Line 164-251) and the revised Supporting Information (Page 19-23). Firstly, we confirmed the layered microstructure of GAF and the block structure of Cu foil based on the cross-sectional image of scanning electron microscope (SEM), and modeled and simulated

their microstructures (**Figure R1**). The electrical boundary conditions are as follows: (1) for the electric field applied in the thickness direction, the lower boundary is grounded (potential of 0 V), and the upper boundary potential is 1 V; (2) for the electric field applied in the in-plane direction, the left boundary is grounded (with a potential of 0 V), and the right boundary has a potential of 4 V. The conductivity of GAF is calculated to be $1.068 \times 10^6 \text{ S m}^{-1}$, which is closed to the real value ($1.07 \times 10^6 \text{ S m}^{-1}$). Solve in frequency domain and derive electric potential distribution results. According to the simulation results, we found that the electric field distribution of GAF in the plane and thickness direction is both consistent with these of Cu foil (**Figure R2**), which meets the application premise of the classical equivalent circuit theory of metal.

Figure R1 The SEM cross-sectional image of (a) GAF and (b) Cu foil.

Figure R2 Electric potential distribution of GAF when applying an electric field in the (a) in-plane direction and (c) thickness direction; electric potential distribution of Cu foil when applying an electric field in the (b) in-plane direction and (d) thickness direction.

Figure R3 (a) Schematic diagram of equivalent circuit model and (b) related parameters of the GAFM.

Then, we designed the GAFM with a rectangular slot structure. In the direction of X or Y polarization, the two sides of the rectangular slot parallel to the polarization axis introduce a capacitive effect, while the GAF conductors on both sides of the rectangular slot parallel to the polarization axis exhibit inductive and resistive effects. Due to the differing length and width of the rectangular structure, the values of resistance (R), inductance (L), and capacitance (C) vary, resulting in distinct R - L - C resonances. This variation enables the polarization shielding manipulation of the GAFM. Additionally, we modeled the GAFM as a combination of capacitance, inductance, and resistance, and calculated its parameters based on the equivalent circuit model (**Figure R3**).

According to transmission line theory and the equivalent circuit, the impedance of GAFM could be expressed as follows:

$$Z_{GAFM} = \frac{(R + j\omega L)}{(1 + j\omega CR - \omega^2 CL)} \quad (1)$$

where ω , R , L , and C are the angular frequency of microwave, equivalent resistance, equivalent inductance, and equivalent capacitance, respectively.

The transmission coefficient (T) of the circular can be calculated by

$$T = \frac{2Z_{GAFM}}{2Z_{GAFM} + Z_0} \quad (2)$$

where Z_0 is the impedance of the free space.

To validate the precision of the proposed equivalent circuit model (ECM) concerning the design parameters of the GAFM, we respectively calculated the response of the equivalent circuit using three different methods: the two-port microwave network mathematical calculation (referred to as "CAL"), the circuit simulation software Advanced Design System (ADS), and the full-wave electromagnetic simulation software CST Microwave Studio. Among them, the parameters of circuit component for the Y polarization direction are as follows: $R = 0.314 \Omega$, $L = 1.550 \text{ nH}$, $C = 0.159 \text{ pF}$, $Z_0 = 377 \Omega$. For the X polarization direction, the parameters of circuit component are as follows: $R = 0.183 \Omega$, $L = 0.162 \text{ nH}$, $C = 0.035 \text{ pF}$, $Z_0 = 377 \Omega$. Based on the results of **Figure R4**, we found that the simulated transmission coefficient of the mathematical model, equivalent circuit model and full wave model are consistent for both Y polarization direction and X polarization direction, verifying that the design of GAFM is feasible from theory to practice. Finally, the field map for the electric field and current conditions of the GAFM under a single polarization direction of the incident electromagnetic wave verifies the above series of results, and proves that the design principle of the dimension of targeted sills is effective (**Figure R5**).

Figure R4 Simulated transmission coefficient of the mathematical model (calculation value, named "CAL"), equivalent circuit model (Advanced Design System, named "ADS") and full wave model (Computer Simulation Technology, named "CST") for (a) Y polarization direction and (b) X polarization direction.

Figure R5 Field map of electric field distribution of GAFM for (a) Y-polarized and (b) X-polarized electromagnetic waves at 10 GHz. Field map of surface current distribution of GAFM for (c) Y-polarized and (d) X-polarized electromagnetic waves at 10 GHz.

We verified **the feasibility of transitioning from a microscopic structural model of carbon materials to the macroscopic electromagnetic equivalent circuit model**. This shift not only facilitates the manipulation of polarization shielding, but also opens the door to a deeper understanding of the material properties at the macroscopic scale, showing the great potential of **extending from metal to carbon material to become the next generation electromagnetic wave response manipulation material**.

Original comment 1-3:

What type of CF film has been used in this study? What's the electrical conductivity of the CF films?

REPLY:

Thanks for your comment. As your comment, we have supplemented the descriptions about CF film. The CF film was prepared from poly-acrylonitrile (PAN) as precursor. PAN fiber was pre-oxidized in air at 300 °C. Then the pre-oxidized fiber was carbonized in inert argon atmosphere at high temperature of 1200 °C, and then introduced into release paper coated with epoxy resin for rolling, and finally the CF film was obtained. Among them, a single carbon fiber has a diameter of about 7 μm and a density of 1.8 g cm⁻³. The electrical conductivity of the CF films is 589 ± 95 S m⁻¹.

The corresponding descriptions have supplemented in the revised manuscript (Page 22, Line 418-423).

Original comment 1-4:

Please provide the unit cell parameter optimization results in the supporting information for analyzing the size effect on the EMI SE.

REPLY:

Thanks for providing valuable insights that will enhance our work. The design and simulation of the GAFM was performed using CST Microwave Studio. The unit cell structure is simulated with periodic boundary conditions along the x and y axes, while Floquet port excitations are applied along the z direction, both above and below the unit cell. **Figure R6a-6b** show the dependence of the EMI SE in Y polarization and X polarization on the length of sl at 7-13 GHz when the slot width is fixed, while **Figure R6c-6d** show the dependence of the EMI SE in Y polarization and X polarization on the length of sw at 7-13 GHz when the slot length is fixed. Therefore, the EMI SE is almost only manipulated by the size parameters parallel to the polarization direction, which proves the anisotropy of the electromagnetic shielding of this structure. At the same time, the manipulation and design of the EMI SE under specific polarization can be realized through changing the structural size parameters. The corresponding descriptions have supplemented in the revised manuscript (Page 11-12, Line 204-239).

Figure R6 Dependence of Δ EMI SE in Y polarization (a) and X polarizations (b) on the lengths of sl at 7-13 GHz ($sw = 17.0$ mm). Dependence of Δ EMI SE in Y polarization (c) and X polarizations (d) on the lengths of sw at 7-13 GHz ($sl = 3.5$ mm).

Original comment 1-5:

Figure 3(f) gives the $\Delta EMI SE$ and $\Delta EMI SE/t$ comparison of different strategies. Please add results of other rotational modulation results as well.

REPLY:

Thanks for your constructive comment. As your suggestion, we have supplemented the references [R1-R3] on the results of rotational manipulation in recent years to Figure 4(f) in the revised manuscript, as shown in **Figure R7**. Notably, when compared to various intelligent EMI shielding materials reported in the references (Details in Table S2), the $\Delta EMI SE/t$ of our designed and prepared GAFM reaches a record-breaking value of 1061.60 dB/mm, while the $\Delta EMI SE$ of GAFM/CF attains a record-high value of 54.33 dB.

Figure R7 (a) Comparison of $\Delta EMI SE$ and $\Delta EMI SE/t$ using the strategy of rotation angle change reported in the references. (b) Comparison of $\Delta EMI SE$ and $\Delta EMI SE/t$ using different strategies (electrochemical potential change, rotation angle change, temperature change, humidity change and shape change) reported in the references.

References

- [R1] Dai, Z. *et al.* Highly anisotropic carbonized wood as electronic materials for electromagnetic interference shielding and thermal management. *Adv. Electron. Mater.* **9**, 2300162 (2023).
- [R2] Xu, L. *et al.* Ultrathin, ultralight, and anisotropic ordered reduced graphene oxide fiber electromagnetic interference shielding membrane. *Adv. Mater. Technol.* **6**, 2100531 (2021).

[R3] Li, D. *et al.* 3D-structured carbon nanotube fibers as ultra-robust fabrics for adaptive electromagnetic shielding. *Nano Res.* **17**, 8521–8530 (2024).

We have replaced Figure 4(f) in the revised manuscript (Page 14) with **Figure R7b**.

Moreover, we have added references [R1-R3] to Table S2 in the revised Supporting Information (Page 28-32).

“Table S2. Comparison of Δ EMI SE and Δ EMI SE/t using different strategies (shape change, temperature change, humidity change, electrochemical potential change, and rotation angle change) reported in the references at X band (8.2 - 12.4 GHz).

No.	Type	Materials	Δ EMI SE (dB)	Δ EMI SE/t (dB/mm)	References
1	Shape change (compression)	Wood-derived carbon/XC-72 NP aerogel	24.00	0.34	S4
2	Shape change (compression)	TPI-MXene/carbon foam	10.00	1.00	S5
3	Shape change (compression)	EVA@PPy@Ag foam	45.00	22.50	S6
4	Shape change (compression)	MF@MXene/Ag NW sponges/PEG	18.10	9.05	S7
5	Shape change (compression)	WF/CNTs composite foam	26.20	5.24	S8
6	Shape change (compression)	PU/CNTs/TPI	11.00	1.10	S9
7	Shape change (compression)	MF@Ag/PDA/CNTs/waterborne polyurethane	9.90	15.00	S10
8	Shape change (compression)	Identical patterned anisotropic magnetic liquid metal/PDMS	33.00	21.88	S11
9	Shape change (compression)	PU/RGO foam-5%	8.00	0.13	S12

10	Shape change (compression)	PU/RGO foam-10%	20.00	0.33	S12
11	Shape change (stretching)	Pre-linked Ni chains elastomer	20.00	66.67	S13
12	Shape change (stretching)	CNT@liquid metal/polyacrylamide/gelatin	37.00	9.25	S14
13	Shape change (stretching)	Fe/liquid metal/PDMS	40.10	50.13	S15
14	Shape change (stretching)	3D liquid metal network	51.70	32.31	S16
15	Shape change (stretching)	Liquid metal elastomer composite (BiInSn)	10.00	2.78	S17
16	Shape change (stretching)	Liquid metal elastomer composite (Ga)	16.00	4.44	S17
17	Shape change (stretching)	Liquid metal foamed elastomer composite (Ga)	25.00	6.94	S17
18	Shape change (stretching)	Ag NPs/SEBS	27.00	10.38	S18
19	Shape change (stretching)	CNT/TPU	22.10	11.05	S19
20	Shape change (stretching)	Liquid metal GaIn _{24.5} /Ni	10.00	200.00	S20
21	Shape change (stretching)	PEDOT:PSS/waterborne PU	4.00	26.67	S21
22	Temperature change	VO ₂ /CNF	34.10	17.95	S22
23	Temperature change	VO ₂ /EPM composites foam	3.00	0.86	S23
24	Temperature change	VO ₂ / PVDF-HFP	44.80	24.89	S24
25	Temperature change	Ti ₃ C ₂ T _x -WVO ₂	13.90	6.95	S25

26	Temperature change	RGO/VO ₂ -300°C	7.00	2.48	S26
27	Temperature change	RGO/VO ₂	46.00	7.93	S26
28	Temperature change	Graphene/PDMS	9.00	1.90	S27
29	Humidity change	Core-shell structural PNIPAM@p-PDA biomicrospheres	61.00	508.33	S28
30	Humidity change	Pyrolytic graphite-wet RGO/CNTs/PP non-woven spacer-pyrolytic graphite	22.00	11.00	S29
31	Electrochemical potential change	MXene film (Ti ₃ C ₂ T _x electrode in 1 M H ₂ SO ₄)/PET	2.20	21.57	S30
32	Electrochemical potential change	MXene film (V ₂ CT _x electrode in 1 M H ₂ SO ₄)/PET	6.00	59.64	S30
33	Rotation angle change	MXene/CNF aerogels	26.00	13.00	S31
34	Rotation angle change	MXene@wood	30.20	15.10	S32
35	Rotation angle change	Polyvinyl butyral/Ni-graphene/short-cut CF composite films	11.00	55.00	S33
36	Rotation angle change	CF reinforced polymer	39.00	70.91	S34
37	Rotation angle change	CF hybrid fabrics	4.50	4.37	S35
38	Rotation angle change	RGO@directional porous carbon	9.05	6.03	S36
39	Rotation angle change	CNT/NFC	38.50	4.81	S37
40	Rotation angle change	Co based amorphous wires	24.10	189.76	S38
41	Rotation angle change	MXene@PDA	17.00	566.67	S39

42	Rotation angle change	Balsa tangential-section carbonized wood	50.00	15.15	S40
43	Rotation angle change	Ordered RGO fiber film	26.00	866.67	S41
44	Rotation angle change	CNT-based spacer fabric with chemical vapor deposition	44.00	88.00	S42
45	Rotation angle change	GAFM	26.54	1061.60	This work
46	Rotation angle change	GAFM/CF	54.33	196.85	This work

Note: nanoparticles (NP), trans-1,4-polyisoprene (TPI), ethylene–vinyl acetate copolymer (EVA), polypyrrole (PPy), silver (Ag), melamine foam (MF), nanowire (NW), poly(ethylene glycol) (PEG), wheat flour (WF), carbon nanotube (CNT), polyurethane (PU), poly-dopamine (PDA), polydimethylsiloxane (PDMS), reduced graphene oxide (RGO), 3 dimension (3D), styrene-(ethylenebutylene)-styrene (SEBS), thermoplastic polyurethane (TPU), cellulose nanofibrils (CNF), Poly(3,4-ethylenedioxythiophene)/poly(styrenesulfonate) (PEDOT:PSS), carbon fiber (CF), vanadium dioxide (VO₂), expanded polymer microsphere (EPM), poly(N-isopropylacrylamide)@porous polydopamine (PNIPAM@p-PDA), polypropylene (PP), poly(ethylene terephthalate) (PET), nano-brillated cellulose (NFC).

The thickness of the sample includes the thickness of the substrate.”

Original comment 1-6:

In Figure 4, except for the schematic diagram, please add the CST simulation diagram to strengthen the mechanism analysis.

REPLY:

Thank you very much for your precious advice. **Figure R8** is the schematic diagram of simulation process. After CST basic setting (frequency, length unit, etc.), GAFM model establishment, periodic boundary conditions and frequency domain solver, and X/Y polarized incident plane wave, shielding results are finally obtained. The simulation results (**Figure R9**) show a vector electric field distribution

of incident electromagnetic wave and compares the EMI SE described by electric field amplitudes ($|\mathbf{E}|$) under X polarization (defined as when the incident electric field is oriented parallel to the long side of the slit) and Y polarization (defined as when the incident electric field is oriented parallel to the short side of the slit), respectively. For Y polarization of the electric field, $|\mathbf{E}|$ remains virtually unchanged both before and after passing through GAFM, which means the undiminished electric field strength and electromagnetic energy. Conversely, for X polarization of the electric field, $|\mathbf{E}|$ transitions from a strong state to an extremely weak disappearing state before and after traversing the GAFM, indicating that most of the electromagnetic energy is efficiently shielded. In addition, in order to further strengthen the mechanism analysis, we have redrawn Figure 5 of the revised manuscript (Page 9-13; Page 17) to ensure a clear explanation of the principle and design mechanism.

Figure R8 Schematic diagram of CST simulation process.

Figure R9 Vector electric field of incident electromagnetic wave and simulated electric field distributions before and after GAFM shielding in X and Y polarizations of electric field, respectively.

Figure R10 Electric field distribution of GAFM for (a) Y-polarized and (b) X-polarized electromagnetic waves at 10 GHz. Surface current distribution of GAFM for (c) Y-polarized and (d) X-polarized electromagnetic waves at 10 GHz. (e) Equivalent circuit of the GAFM.

Figure R10a and R10b depict the electric field distribution for incident Y-polarized and X-polarized electromagnetic waves at 10 GHz, respectively, whereas **Figure R10c and R10d** illustrate the corresponding surface current distributions under the same polarization conditions. A significant disparity is observed in the magnitude of the electric field and current between the X and Y polarization directions. The designed structure functions as a spatial band-pass filter, exhibiting distinct electromagnetic response characteristics in the X and Y directions attributable to its anisotropic properties. For incident Y-polarized electromagnetic waves, the band-pass resonance point is precisely located within the X-band, thereby demonstrating clear band-pass characteristics. Conversely, for incident X-polarized electromagnetic waves, the relatively short electrical length of the structure shifts the band-pass response frequency to higher frequencies, resulting in band-stop characteristics for electromagnetic waves located outside the passband. **Figure R10e** is the corresponding equivalent circuit of the GAFM.

The corresponding descriptions have supplemented in the revised manuscript (Page 9-13, Line 164-251; Page 17, Line 288-312) and the revised Supporting Information (Page 19-23).

Response to Reviewer #2

REPLY:

We would like to sincerely thank you for the considerable time and effort you have dedicated to helping improve the presentation of our manuscript. Your preciseness, patience, and attention to detail have significantly enhanced both the quality and scientific rigor of the paper. We greatly appreciate your constructive comments and suggestions, which have been instrumental in refining our work. We hope that the revised manuscript meets your expectations. To ensure a comprehensive response, we have addressed each of your comments one by one.

Original comment 2-1:

The unit cell dimension is 18 mm, which is around $\lambda/2$ in comparison with the operating band. This is a very well-known fact, there is no metamaterial.

REPLY:

Thank you for this comment. Indeed, in the traditional concept, as you said, the special electromagnetic structure whose periodic unit size is much smaller than the working wavelength is called metamaterial. However, in a broad sense, the periodic structure that can give electromagnetic materials beyond the properties of materials in nature can be called metamaterials. Now more sub-wavelength (the size is equal to or smaller than the working wavelength) is used to define metamaterials ^[R1-R3].

In this work, we designed the geometric structure of GAF materials based on electromagnetic theory, and transformed the isotropic GAF in plane into anisotropic GAFM in plane by constructing slits with specific structures, thus endowing it with polarization sensitive properties, which is a characteristic that natural materials do not have. Specifically, the EMI SE is lowest when the incident electric field is oriented parallel to the short side of the slit, while the EMI SE is highest when the incident electric field polarization is orthogonal to the short side of the slit. Therefore, when the angle between the slit orientation (the long side direction) in GAFM and the direction of the electric field is rotated from 0° to 90° , the EMI SE changes from maximum value (almost completely shielding) to minimum value (almost entirely transmitting). Therefore, we believe GAFM is a type of metamaterial.

In addition, in Figure S9 in the revised Supporting Information (Page 20), we also demonstrated a periodic structure whose unit size is much smaller than the working wavelength, which can also realize polarization manipulation, but the switch ratio is worse than the size used in the manuscript. In a broad definition, this series can all be called metamaterials. At the same time, in order to better illustrate this point, we have made a detailed explanation in the revised manuscript (Page 5-6, Line 99-108) and the revised Supporting Information (Page 4-6, 19).

References

[R1] Cheng, Y. Z. *et al.* Design, fabrication and measurement of a broadband polarization-insensitive metamaterial absorber based on lumped elements. *J. Appl. Phys.* **111**, 044902 (2012).

[R2] Tuong, P. V, Park, J. W., Kim, Y. J., Yoo, Y. J. & Lee, Y. P. Broadband Reflection of Polarization Conversion by 90 ° in Metamaterial. *J. Korean Phys. Soc.* **64**, 1116–1119 (2014).

[R3] Zhang, C. *et al.* Broadband metamaterial for optical transparency and microwave absorption. *Appl. Phys. Lett.* **110**, 143511 (2017).

Original comment 2-2:

Further, this may invoke non-specular modes which must be investigated by the authors.

REPLY:

We sincerely thank you for this valuable advice. To effectively analyze the non-specular reflection modes of the GAFM, as illustrated in **Figure R11**, we conducted simulations using CST to examine the far-field radiation patterns of a 10×10 GAFM array. Plane electromagnetic waves polarized in the Y and X directions were used as excitation sources. Under conditions of normal incidence, the primary reflected beam of the GAFM array aligns perpendicularly with the incident direction, thereby demonstrating the absence of non-specular reflection phenomena. Additionally, at 10 GHz, the intensity of the reflected beam for Y-polarized waves is significantly lower than at 8 GHz and 12 GHz, while X-polarized waves consistently exhibit higher amplitude reflections. These results align with individual element simulations.

Figure R11 3D far-field radiation patterns of the GAFM at 8 GHz, 10 GHz, and 12 GHz for plane waves with (a-c) Y-polarization and (d-f) X-polarization.

For further analysis, a horn antenna was used as the feed source in simulations. **Figure R12a-12c** show the far-field radiation patterns of the horn antenna, and **Figure R12d-12i** depict the far-field beams of the GAFM array with the horn. The results obtained from horn feeds with different polarization orientations are consistent with those derived from plane wave simulation, confirming the reliability of our approach.

Figure R12 3D far-field radiation patterns of Y-polarized horn antennas at (a) 8 GHz, (b) 10 GHz, and (c) 12 GHz. 3D far-field radiation patterns of the GAFM employing horn antennas as excitation sources at (d) 8 GHz, (e) 10 GHz, and (f) 12 GHz with Y-polarization; and the corresponding patterns at (g) 8 GHz, (h) 10 GHz, and (i) 12 GHz with X-polarization.

The corresponding descriptions have been supplemented in the revised manuscript (Page 11-13, Line 218-251) and the revised Supporting Information (Page 21-23).

Original comment 2-3:

What is the role of graphene in the design? What will happen if the graphene is replaced by conventional metal or dielectric?

REPLY:

Thank you for this comment. In this work, we don't just report the design of EMI SE based on graphene materials, but take graphene assembly film materials and graphene assembly film/carbon fiber materials as representatives, and demonstrate a strategy of popularizing metal materials to advanced carbon materials in the scene of polarization manipulation of electromagnetic shielding interference effectiveness and full polarization shielding.

With the rapid development of information technology, electromagnetic materials and devices have been widely used in aerospace, military equipment, wireless communication and other strategic fields. "Lightweight, flexibility and long life" has become an urgent demand for the next generation of electromagnetic materials and devices. However, the most widely used metal materials are gradually unable to meet the requirements because of their high density, poor bending resistance and short life, etc.

Compared with conventional metals, carbon materials (especially graphene) have the following advantages: ^[R4-R13]

Low density: The density of graphene assembled film is not more than 2.2 g cm^{-3} , which is much lower than that of copper (8.96 g cm^{-3}). Low density can reduce the overall weight, reduce energy consumption, and improve portability and operational efficiency in lightweight applications.

Good fatigue resistance: Graphene has excellent fatigue resistance, while metal materials are prone to fatigue cracks in the long-term use process, leading to performance degradation.

Strong corrosion resistance: Carbon materials exhibit superior resistance to both acidic and alkaline corrosion, in contrast to metals, which show relatively poor resistance in these environments.

Excellent electrical conductivity: Graphene has high electrical conductivity of 10^6 S m^{-1} , making it the best non-metallic material at present and achieving similar performance to metal based devices. ^[R6]

Therefore, although metal materials can also achieve similar designs in this manuscript, carbon materials will be the materials with great application potential in the future. This manuscript aims to provide an important reference for the subsequent application and popularization of other carbon materials at the metal-level manipulation of electromagnetic interference shielding effectiveness.

As for dielectric materials, they often have low conductivity, so it is difficult to form induced current for incident electromagnetic waves in electromagnetic fields, and thus electromagnetic energy can only be

dissipated in the form of heat loss. In this way, if the graphene is replaced by dielectric materials, the polarization manipulation of electromagnetic wave response cannot be realized.

References

- [R4] Novoselov, K. S. *et al.* A roadmap for graphene. *Nature* **490**, 192–200 (2012).
- [R5] Yu, X. *et al.* Graphene-based smart materials. *Nat. Rev. Mater.* **2**, 1–14 (2017).
- [R6] Song, R. *et al.* Comparison of copper and graphene-assembled films in 5G wireless communication and THz electromagnetic-interference shielding. *Proc. Natl. Acad. Sci.* **120**, e2209807120 (2023).
- [R7] Kong, W. *et al.* Path towards graphene commercialization from lab to market. *Nat. Nanotechnol.* **14**, 927–938 (2019).
- [R8] Lundberg, M. B. *et al.* Tuning quantum nonlocal effects in graphene plasmonics. *Science* **357**, 187–191 (2017).
- [R9] Li, L. *et al.* Large-scale current collectors for regulating heat transfer and enhancing battery safety. *Nat. Chem. Eng.* **1**, 542–551 (2024).
- [R10] Wang, S. *et al.* Gate-tunable plasmons in mixed-dimensional van der Waals heterostructures. *Nat. Commun.* **12**, 5039 (2021).
- [R11] Liu, M. *et al.* A popcorn-inspired strategy for compounding graphene@NiFe₂O₄ flexible films for strong electromagnetic interference shielding and absorption. *Nat. Commun.* **15**, 5486 (2024).
- [R12] Lv, H. *et al.* Staggered circular nanoporous graphene converts electromagnetic waves into electricity. *Nat. Commun.* **14**, 1982 (2023).
- [R13] Geim, A. K. & Novoselov, K. S. The rise of graphene. *Nat. Mater.* **6**, 183–191 (2007).

Original comment 2-4:

How the graphene layer has been prepared? How they have been modelled theoretically?

REPLY:

Thank you for this valuable comment.

The formation of the graphene layer derives from the reduction of graphene oxide (GO) layer, which is closely related to the preparation process of graphene assembly film (GAF) as shown in **Figure R13**. Firstly, the precursor GO is uniformly dispersed in ultrapure water to form a slurry, then GO nanosheets are assembled into a GO film through coating and drying. After further high-temperature heat treatment, reduction and defect repair are achieved, and finally GAF is obtained through rolling compression process. The conductivity of GAF reaches $1.07 \times 10^6 \text{ S m}^{-1}$, which is a breakthrough in carbon-based materials.

Figure R13 The fabrication process of GAF. (a) Schematic diagram of GAF preparation process. Images of (b) coating, (c) annealing furnace, and (d) rolling compression.

We apologize for the insufficient detail in the theoretical modeling section of graphene materials in our previous manuscript and have provided additional information to address this.

As illustrated in **Figure R14**, the GAF with $25 \mu\text{m}$ thickness is composed of many stacked graphene layers. Based on the layer stacking structure of GAF in the cross-sectional SEM image and ultrahigh conductivity, we established a microscopic model (**Figure R15**). Among them, the layered structure simulates the microstructure of GAF, while the rectangular blocks represent the Cu metal blocks as the control group. The electrical boundary conditions are as follows: (1) for the electric field applied in the thickness direction, the lower boundary is grounded (potential of 0 V), and the upper boundary potential is 1 V; (2) for the electric field applied in the in-plane direction, the left boundary is grounded (with a potential of 0 V), and the right boundary has a potential of 4 V. The conductivity of GAF is calculated to be $1.068 \times 10^6 \text{ S m}^{-1}$, which is closed to the real value ($1.07 \times 10^6 \text{ S m}^{-1}$). Solve in frequency domain and

derive electric potential distribution results. The simulation results show that regardless of whether the direction of the applied electric field is the thickness direction or the plane direction, GAF exhibits similar characteristics to Cu metal in terms of microstructure response, which meets the application premise of the classical equivalent circuit theory of metal.

Figure R14 (a) Thickness of 25 μm and (b) SEM cross-sectional image of GAF. (c) SEM cross-sectional image of Cu foil.

Figure R15 Electric potential distribution of GAF when applying an electric field in the (a) in-plane direction and (c) thickness direction; electric potential distribution of Cu foil when applying an electric field in the (b) in-plane direction and (d) thickness direction.

Then, we modeled the GAFM as a combination of capacitance-inductance-resistance and calculated parameters of GAFM based on equivalent circuit model (**Figure R16**).

Figure R16 (a) Schematic diagram of equivalent circuit model and (b) related parameters of the GAFM.

According to transmission line theory and the equivalent circuit, the impedance of GAFM could be expressed as follows:

$$Z_{GAFM} = \frac{(R + j\omega L)}{(1 + j\omega CR - \omega^2 CL)} \quad (1)$$

where ω , R , L , and C are the angular frequency of microwave, equivalent resistance, equivalent inductance, and equivalent capacitance, respectively.

The transmission coefficient (T) of the circular can be calculated by

$$T = \frac{2Z_{GAFM}}{2Z_{GAFM} + Z_0} \quad (2)$$

where Z_0 is the impedance of the free space.

To validate the precision of the proposed equivalent circuit model (ECM) concerning the design parameters of the GAFM, we respectively calculated the response of the equivalent circuit using three different methods: the two-port microwave network mathematical calculation (referred to as "CAL"), the circuit simulation software Advanced Design System (ADS), and the full-wave electromagnetic simulation software CST Microwave Studio. Among them, the parameters of circuit component for the Y polarization direction are as follows: $R = 0.314 \Omega$, $L = 1.550 \text{ nH}$, $C = 0.159 \text{ pF}$, $Z_0 = 377 \Omega$. For the X polarization direction, the parameters of circuit component are as follows: $R = 0.183 \Omega$, $L = 0.162 \text{ nH}$, C

$= 0.035 \text{ pF}$, $Z_0 = 377 \ \Omega$. Based on the results of **Figure R17**, we found that the simulated transmission coefficient of the mathematical model, equivalent circuit model and full wave model are consistent for both Y polarization direction and X polarization direction, verifying that the design of GAFM is feasible from theory to practice.

Figure R17 Simulated transmission coefficient of the mathematical model (calculation value, named “CAL”), equivalent circuit model (Advanced Design System, named “ADS”) and full wave model (Computer Simulation Technology, named “CST”) for (a) Y polarization direction and (b) X polarization direction.

Finally, the field map for the electric field and current conditions of the GAFM under a single polarization direction of the incident electromagnetic wave verifies the above series of results, and proves that the design principle of the dimension of targeted sills is effective (**Figure R18**).

Figure R18 Field map of electric field distribution of GAFM for (a) Y-polarized and (b) X-polarized electromagnetic waves at 10 GHz. Field map of surface current distribution of GAFM for (c) Y-polarized and (d) X-polarized electromagnetic waves at 10 GHz.

We verified **the feasibility of transitioning from a microscopic structural model of carbon materials to the macroscopic electromagnetic equivalent circuit model**. This shift not only facilitates the manipulation of polarization shielding, but also opens the door to a deeper understanding of the material properties at the macroscopic scale. It can be seen that this multi-layer carbon material structure with ultra-high conductivity shows the great potential of **extending from metal to carbon material to become the next generation electromagnetic wave response manipulation material**.

We have added relevant descriptions in the revised manuscript (Page 9-13, Line 164-251) and the revised Supporting Information (Page 7-9).

Original comment 2-5:

The experimental frequency responses have not been shown.

REPLY:

We greatly appreciate you for this insightful suggestion. Benefiting from your well-intentioned guidance, we have shown the experimental frequency responses of GAFM (**Figure R19a**), CF (**Figure R19b**), and GAFM/CF (**Figure R20**) including GAFM // CF and GAFM \perp CF two modes in the revised manuscript (Page 13-14, Line 252-271) and the revised Supporting Information (Page 18).

Figure R19 The EMI SE of (a) GAFM and (b) CF when the rotation angle is from 0° to 90°.

When the rotating angle is 0° (defined as when the incident electric field is oriented parallel to the short side of the slit), the EMI SE of GAFM diminishes to a minimum of 0.14 dB (shielding efficiency of 3.17%), closely approximating 0 dB and reaching the lowest value reported in references, thus facilitating efficient ‘on and off’ manipulation (**Figure R19a**). Conversely, when the rotating angle is 90° (defined as when the incident electric field is oriented parallel to the long side of the slit), the EMI SE of GAFM

consistently maintains above 20 dB over the entire frequency band, peaking at 26.68 dB, thereby indicating a maximum shielding efficiency of 99.79%. We define the direction of CF as 0° when the CF orientation is perpendicular to the incident electric field polarization. With the gradual increase from 0° to 90° in rotation angle, the EMI SE of CF gradually improves, as depicted in **Figure R19b**.

To achieve expanded polarization difference range and fully polarized shielding, we employed two corresponding modes: parallel combination (GAFM//CF, **Figure R20a-R20b**) and vertical combination (GAFM \perp CF, **Figure R20c-R20d**), respectively. Upon increasing the rotation angle of GAFM//CF from 0° to 90° , and subsequently to 180° , the EMI SE within the X band shows a trend of initial increase followed by a decrease. The polarization shielding difference is notably enhanced to 54.33 dB, which represents a significant enhancement in polarization difference. Conversely, when the rotation angle of GAFM \perp CF increases from 0° to 90° , and further to 180° , the EMI SE within the X band demonstrates a trend of initial decrease followed by an increase. Most importantly, the GAFM \perp CF case achieves a smaller EMI SE difference with the full-angle EMI SE exceeding 20.7 dB (full-angle EMI shielding over 99.15%), indicating fully polarized EMI shielding.

Figure R20 The EMI SE of (a,b) GAFM//CF and (c,d) GAFM \perp CF when the rotation angle is from 0° to 180° .

Original comment 2-6:

How the structure is advantageous to the reported ones?

REPLY:

Thank you for this comment. We are sorry that we have not clarified the highlights of this work better. In fact, the core contributions of this work go beyond the structural design of a metamaterial with anisotropic properties. The real innovation lies in our extension of classical electromagnetic equivalent circuit theory—traditionally applied to metallic materials—to advanced carbon materials with microstructures, such as graphene-assembled films (GAF). This shift in theoretical approach allows us to not only elucidate the EMI shielding mechanism and polarization manipulation mechanism of advanced carbon materials, but also develop serialized EMI SE manipulation metamaterials with some record-breaking performances—specifically, the widest polarization difference manipulation range, the highest full polarization shielding and the most significant polarization switching effect (Figure R20), simultaneously. This is a major departure from previous work on EMI SE manipulation with carbon materials, which has lacked a strong theoretical foundation and has often resulted in lower performance.

Figure R21 Comparison of initial EMI shielding efficiency, manipulated EMI shielding efficiency, Δ EMI shielding efficiency using different strategies (shape change, temperature change, humidity change, electrochemical potential change, and rotation angle change) reported in the references at X band (8.2 - 12.4 GHz).

However, it is worth noting that not all carbon materials can achieve this effect, and the following two conditions must be met: (1) **the electrical conductivity needs to be high enough**, and (2) **the microscopic cross-sectional structure can be consistent with the characteristics of metals in both in-plane and out-of-plane directions**.

For the first point, the GAF we prepared possesses high electrical conductivity of $1.07 \times 10^6 \text{ S m}^{-1}$, and can generate enough induced current for incident electromagnetic waves to realize attenuation of electromagnetic waves.

For the second point, we established a microscopic model based on the structure of graphene layer stacking and confirmed that the microscopic cross-sectional structure of GAF is consistent with the characteristics of metals in both in-plane and out-of-plane directions (**Figure R15**).

In addition, the design of this work is not limited to GAF, but also includes the joint design of GAF and CF. The anisotropy of these two materials is obtained by artificial manufacturing, which not only realizes accurate manipulation of EMI SE, but also realizes the **highest full polarization shielding**.

The corresponding summary has supplemented in the revised Supporting Information (Page 13-17) as follows:

“Table S1. Comparison of initial EMI shielding efficiency, manipulated EMI shielding efficiency, Δ EMI shielding efficiency using different strategies (shape change, temperature change, humidity change, electrochemical potential change, and rotation angle change) reported in the references at X band (8.2 - 12.4 GHz).

No	Type	Materials	Initial EMI shielding efficiency (%)	Manipulated EMI shielding efficiency (%)	Δ EMI shielding efficiency (%)	References
1	Shape change (compression)	Wood-derived carbon/XC-72 NP aerogel	29.2054215616	99.7181617069	70.5127401453	S4

2	Shape change (compression)	TPI-MXene/carbon foam	99.6837722 340	96.8377223 398	2.8460498942	S5
3	Shape change (compression)	EVA@PPy@Ag foam	99.9999308 169	76.1768053 064	23.8231255105	S6
4	Shape change (compression)	MF@MXene/Ag NW sponges/PEG	94.2456006 266	99.9108749 062	5.6652742796	S7
5	Shape change (compression)	WF/CNTs composite foam	99.9902276 278	95.9261972 220	4.0640304058	S8
6	Shape change (compression)	PU/CNTs/TPI	99.9668868 879	99.5831306 165	0.3837562714	S9
7	Shape change (compression)	MF@Ag/PDA/CNTs /waterborne polyurethane	99.9754529 108	99.6980048 280	0.2774480828	S10
8	Shape change (compression)	Identical patterned anisotropic magnetic liquid metal/PDMS	87.4107458 821	99.9936904 266	12.5829445445	S11
9	Shape change (compression)	PU/RGO foam-5%	99.4988127 664	96.8377223 398	2.6610904266	S12
10	Shape change (compression)	PU/RGO foam-10%	99.9936904 266	99.3690426 555	0.6246477711	S12
11	Shape change (stretching)	Pre-linked Ni chains elastomer	36.9042655 520	99.3690426 555	62.4647771035	S13
12	Shape change (stretching)	CNT@liquid metal/polyacrylamide /gelatin	99.9900000 000	49.8812766 373	50.1087233627	S14
13	Shape change (stretching)	Fe/liquid metal/PDMS	99.1290364 100	99.9999991 489	0.8709627389	S15
14	Shape change (stretching)	3D liquid metal network	99.9999997 601	99.9645186 611	0.0354810990	S16
15	Shape change (stretching)	Liquid metal elastomer composite (BiInSn)	92.0567176 528	20.5671765 276	71.4895411252	S17
16	Shape change (stretching)	Liquid metal elastomer composite (Ga)	92.0567176 528	99.8004737 685	7.7437561157	S17

17	Shape change (stretching)	Liquid metal foamed elastomer composite (Ga)	99.9999000 000	99.9999996 838	0.0000996838	S17
18	Shape change (stretching)	Ag NPs/SEBS	99.9996837 722	99.8415106 808	0.1581730914	S18
19	Shape change (stretching)	CNT/TPU	99.9653263 150	94.3765867 481	5.5887395669	S19
20	Shape change (stretching)	Liquid metal GaIn _{24.5} /Ni	99.9999900 000	99.9999000 000	0.0000900000	S20
21	Shape change (stretching)	PEDOT:PSS/waterborne PU	99.9998415 107	99.9996018 928	0.0002396179	S21
22	Temperature change	VO ₂ /CNF	98.6510371 174	99.9994751 925	1.3484380751	S22
23	Temperature change	VO ₂ /EPM composites foam	52.6848741 039	5.59391237 140	47.0909617325	S23
24	Temperature change	VO ₂ /PVDF-HFP	80.0473768 503	99.9993393 066	19.9519624563	S24
25	Temperature change	Ti ₃ C ₂ T _x -WVO ₂	99.8711750 448	99.9947519 254	0.1235768806	S25
26	Temperature change	RGO/VO ₂ -300°C	60.1892829 447	92.0567176 528	31.8674347081	S26
27	Temperature change	RGO/VO ₂	84.1510680 754	99.9996018 928	15.8485338174	S26
28	Temperature change	Graphene/PDMS	99.9910874 906	99.9205671 765	0.0705203141	S27
29	Humidity change	Core-shell structural PNIPAM@p-PDA biomicrospheres	99.9601892 829	99.9999999 684	0.0398106855	S28
30	Humidity change	Pyrolytic graphite-wet RGO/CNTs/PP non-woven spacer-pyrolytic graphite	97.4881135 685	99.9841510 681	2.4960374996	S29
31	Electrochemical potential change	MXene film (Ti ₃ C ₂ T _x electrode in 1 M H ₂ SO ₄)/PET	99.9205671 765	99.9521369 908	0.0315698143	S30

32	Electrochemical potential change	MXene film (V_2CT_x electrode in 1 M H_2SO_4)/PET	96.9800482796	99.2414224250	2.2613741454	S30
33	Rotation angle change	MXene/CNF aerogels	99.9999498813	99.9800473769	0.0199025044	S31
34	Rotation angle change	MXene@wood	99.9998262199	99.8180299141	0.1817963058	S32
35	Rotation angle change	Polyvinyl butyral/Ni-graphene/short-cut CF composite films	99.0000000000	99.9205671765	0.9205671765	S33
36	Rotation angle change	CF reinforced polymer	99.9987410746	90.0000000000	9.9987410746	S34
37	Rotation angle change	CF hybrid fabrics	43.7658674810	80.0473768503	36.2815093693	S35
38	Rotation angle change	RGO@directional porous carbon	99.6764063431	99.9597282966	0.2833219535	S36
39	Rotation angle change	CNT/NFC	99.9999936904	99.9553316408	0.0446620496	S37
40	Rotation angle change	Co based amorphous wires	99.6611558439	12.9036410044	86.7575148395	S38
41	Rotation angle change	MXene@polydopamine	98.7410745882	36.9042655520	61.8368090362	S39
42	Rotation angle change	Balsa tangential-section carbonized wood	99.9999990000	99.9000000000	0.0999990000	S40
43	Rotation angle change	Ordered RGO fiber film	99.9205671765	68.3772233983	31.5433437782	S41
44	Rotation angle change	CNT-based spacer fabric with chemical vapor deposition	99.9498812766	99.9999998005	0.0501185239	S42
45	Rotation angle change	GAFM	3.17221437388	99.7852169526	96.6130025787	This work

Note: nanoparticles (NP), trans-1,4-polyisoprene (TPI), ethylene–vinyl acetate copolymer (EVA), polypyrrole (PPy), silver (Ag), melamine foam (MF), nanowire (NW), poly(ethylene glycol) (PEG), wheat flour (WF), carbon nanotube (CNT), polyurethane (PU), poly-dopamine (PDA),

polydimethylsiloxane (PDMS), reduced graphene oxide (RGO), 3 dimension (3D), styrene-(ethylenebutylene)-styrene (SEBS), thermoplastic polyurethane (TPU), cellulose nanofibrils (CNF), poly(3,4-ethylenedioxythiophene)/poly(styrenesulfonate) (PEDOT:PSS), carbon fiber (CF), vanadium dioxide (VO₂), expanded polymer microsphere (EPM), poly (vinylidene fluoride-co-hexafluoropropylene (PVDF-HFP), poly(N-isopropylacrylamide)@ porous polydopamine (PNIPAM@p-PDA), polypropylene (PP), poly(ethylene terephthalate) (PET), nano-brillated cellulose (NFC).”

Response to Reviewer #3

General comment:

The paper reports the development of thin EMI shielding film using a highly conducting graphene sheet. The EMI mechanism is altered by making slots in the conducting sheet. Additionally, an oriented CF sheet is used to reflect the EM radiation. The cross structure between graphene slots and the alignment of CF gives the polarization rotation. The paper can not be acceptable in its present form for the following reasons.

REPLY:

First of all, we deeply appreciate you for the plenty of time and effort spent to help us improve the paper. Your comments are very professional and constructive. Based on your criticism, we have thoughtfully reflected on the innovation of this work, added several important experiments and simulations, and almost rewritten the entire paper. Your meritorious help and guidance undoubtedly improved the quality of this paper significantly. Next, we reply to your comments one by one, especially to clarify the most important innovation of this work, and we sincerely hope that the improved paper can meet your requirements.

Original comment 3-1:

Apart from lacking novelty, the work does not quantify many parameters. For example, the relation between the width of the slot and the efficiency of polarization rotation/shielding efficiency etc.

REPLY:

Thank you for this valuable suggestion. We are sorry that we have not clarified the highlights of this work better. The key innovation of this work lies in providing a sort of significant conceptual advance by extending classical electromagnetic equivalent circuit theory, traditionally applied to metallic materials, to advanced carbon materials with microstructures (such as GAF). We have not only developed serialized EMI SE manipulation metamaterials with a record-breaking performance (including the widest polarization difference manipulation range, the highest full polarization shielding and the most significant polarization switching effect) simultaneously based on advanced carbon materials through design, but also put forward and illuminated the shielding mechanism of carbon-based

materials in depth. Furthermore, we verified the feasibility of transitioning from the microscopic structural model of carbon materials to the macroscopic electromagnetic equivalent circuit model, which enabled the manipulative design of polarization shielding in carbon based materials. This is a major departure from previous work on EMI SE manipulation with carbon materials, which has lacked a strong theoretical foundation and has often resulted in lower performance.

Figure R22 (a) Comparison of $\Delta\text{EMI SE}$ and $\Delta\text{EMI SE}/t$ using the strategy of rotation angle change reported in the references. (b) Comparison of $\Delta\text{EMI SE}$ and $\Delta\text{EMI SE}/t$ using different strategies (electrochemical potential change, rotation angle change, temperature change, humidity change and shape change) reported in the references.

As you know, with the rapid development of information technology, electromagnetic materials and devices have been widely used in aerospace, military equipment, wireless communication and other strategic fields. “Lightweight, flexibility and long life” has become an urgent demand for the next generation of electromagnetic materials and devices. However, the most widely used metal materials are gradually unable to meet the requirements because of their high density, poor bending resistance and short life, etc. Compared with conventional metals, carbon materials (especially graphene and carbon fiber) have the following advantages: ^[R14-R23] low density, good fatigue resistance, strong corrosion resistance, excellent electrical conductivity, etc. Therefore, compared with conventional metal materials, carbon materials will be the materials with great application potential in the field of electromagnetism in the future.

Furthermore, this manuscript enables the manipulative design of polarization shielding of carbon-based materials, extending classical electromagnetic equivalent circuit theory—traditionally applied to metallic materials—to advanced carbon materials with microstructures, such as graphene-assembled films (GAF), rather than only the design of structure and shape for GAF.

It is worth noting that **not all carbon materials can achieve this effect**, and the following two conditions must be met. (1) **The electrical conductivity needs to be high enough.** (2) **The microscopic cross-sectional structure can be consistent with the characteristics of metals in both in-plane and out-of-plane directions.**

For the first point, the GAF we prepared possesses high electrical conductivity of $1.07 \times 10^6 \text{ S m}^{-1}$, and can generate enough induced current for incident electromagnetic waves to realize attenuation of electromagnetic waves.

Figure R23 The SEM cross-sectional image of (a) GAF and (b) Cu foil.

Figure R24 Electric potential distribution of GAF when applying an electric field in the (a) in-plane direction and (c) thickness direction; electric potential distribution of Cu foil when applying an electric field in the (b) in-plane direction and (d) thickness direction.

For the second point, we established a microscopic model based on the layer-stacking structure of GAF (**Figure R23**), and confirmed that the electric field distribution of GAF is consistent with that of Cu metals in both in-plane and thickness directions (**Figure R24**). The layered microstructure of GAF was simulated, while the rectangular blocks represent Cu metal blocks as the control group. The electrical boundary conditions are as follows: (1) for the electric field applied in the thickness direction, the lower boundary is grounded (potential of 0 V), and the upper boundary potential is 1 V; (2) for the electric field applied in the in-plane direction, the left boundary is grounded (with a potential of 0 V), and the right boundary has a potential of 4 V. The conductivity of GAF is calculated to be $1.068 \times 10^6 \text{ S m}^{-1}$, which is closed to the real value ($1.07 \times 10^6 \text{ S m}^{-1}$). Solve in frequency domain and derive electric potential distribution results. The simulation results show that regardless of whether the direction of the applied electric field is in the thickness direction or the plane direction, GAF exhibits similar characteristics to metals in terms of microstructure response, which meets the application premise of the classical equivalent circuit theory of metal.

Figure R25 (a) Schematic diagram of equivalent circuit model and (b) related parameters of the GAFM.

Then, we modeled the GAFM as a combination of capacitance-inductance-resistance and calculated parameters of GAFM based on equivalent circuit model (**Figure R25**).

According to transmission line theory and the equivalent circuit, the impedance of GAFM could be expressed as follows:

$$Z_{GAFM} = \frac{(R + j\omega L)}{(1 + j\omega CR - \omega^2 CL)} \quad (1)$$

where ω , R , L , and C are the angular frequency of microwave, equivalent resistance, equivalent inductance, and equivalent capacitance, respectively.

The transmission coefficient (T) of the circular can be calculated by

$$T = \frac{2Z_{GAFM}}{2Z_{GAFM} + Z_0} \quad (2)$$

where Z_0 is the impedance of the free space.

To validate the precision of the proposed equivalent circuit model (ECM) concerning the design parameters of the GAFM, we respectively calculated the response of the equivalent circuit using three different methods: the two-port microwave network mathematical calculation (referred to as "CAL"), the circuit simulation software Advanced Design System (ADS), and the full-wave electromagnetic simulation software CST Microwave Studio. Among them, the parameters of circuit component for the Y polarization direction are as follows: $R = 0.314 \Omega$, $L = 1.550 \text{ nH}$, $C = 0.159 \text{ pF}$, $Z_0 = 377 \Omega$. For the X polarization direction, the parameters of circuit component are as follows: $R = 0.183 \Omega$, $L = 0.162 \text{ nH}$, $C = 0.035 \text{ pF}$, $Z_0 = 377 \Omega$. Based on the results of **Figure R26**, we found that the simulated transmission coefficient of the mathematical model, equivalent circuit model and full wave model are consistent for both Y polarization direction and X polarization direction, verifying that the design of GAFM is feasible from theory to practice.

Finally, the field map for the electric field and current conditions of the GAFM under a single polarization direction of the incident electromagnetic wave verifies the above series of results, and proves that the design principle of the dimension of targeted sills is effective (**Figure R27**).

We verified **the feasibility of transitioning from a microscopic structural model of carbon materials to the macroscopic electromagnetic equivalent circuit model**. This shift not only facilitates the manipulation of polarization shielding, but also opens the door to a deeper understanding of the material properties at the macroscopic scale, showing the great potential of **extending from metal to carbon material to become the next generation electromagnetic wave response manipulation material**.

Figure R26 Simulated transmission coefficient of the mathematical model (calculation value, named “CAL”), equivalent circuit model (Advanced Design System, named “ADS”) and full wave model (Computer Simulation Technology, named “CST”) for (a) Y polarization direction and (b) X polarization direction.

Figure R27 Field map of electric field distribution of GAFM for (a) Y-polarized and (b) X-polarized electromagnetic waves at 10 GHz. Field map of surface current distribution of GAFM for (c) Y-polarized and (d) X-polarized electromagnetic waves at 10 GHz.

In addition, we investigated the effect of slot length and width on electromagnetic shielding effectiveness. The design and simulation of the GAFM was performed using CST Microwave Studio. The unit cell

structure is simulated with periodic boundary conditions along the x and y axes, while Floquet port excitations are applied along the z direction, both above and below the unit cell. **Figure R28a-R28b** shows the dependence of the EMI SE in Y polarization and X polarization on the length of sl at 7-13 GHz when the slot width is fixed, while **Figure R28c-R28d** show the dependence of the EMI SE in Y polarization and X polarization on the length of sw at 7-13 GHz when the slot length is fixed. Therefore, the EMI SE is almost only manipulated by the size parameters parallel to the polarization direction, which proves the anisotropy of the electromagnetic shielding of this structure. At the same time, the manipulation and design of the EMI SE under specific polarization can be realized through changing the structural size parameters.

Figure R28 Dependence of Δ EMI SE in Y polarization (a) and X polarizations (b) on the lengths of sl at 7-13 GHz ($sw = 17.0$ mm). Dependence of Δ EMI SE in Y polarization (c) and X polarizations (d) on the lengths of sw at 7-13 GHz ($sl = 3.5$ mm).

We have added relevant descriptions in the revised manuscript (Page 9-13, Line 164-251) and the revised Supporting Information (Page 19-23).

References

[R14] Novoselov, K. S. *et al.* A roadmap for graphene. *Nature* **490**, 192–200 (2012).

- [R15] Yu, X. *et al.* Graphene-based smart materials. *Nat. Rev. Mater.* **2**, 1–14 (2017).
- [R16] Song, R. *et al.* Comparison of copper and graphene-assembled films in 5G wireless communication and THz electromagnetic-interference shielding. *Proc. Natl. Acad. Sci.* **120**, e2209807120 (2023).
- [R17] Kong, W. *et al.* Path towards graphene commercialization from lab to market. *Nat. Nanotechnol.* **14**, 927–938 (2019).
- [R18] Lundeberg, M. B. *et al.* Tuning quantum nonlocal effects in graphene plasmonics. *Science* **357**, 187–191 (2017).
- [R19] Li, L. *et al.* Large-scale current collectors for regulating heat transfer and enhancing battery safety. *Nat. Chem. Eng.* **1**, 542–551 (2024).
- [R20] Wang, S. *et al.* Gate-tunable plasmons in mixed-dimensional van der Waals heterostructures. *Nat. Commun.* **12**, 5039 (2021).
- [R21] Liu, M. *et al.* A popcorn-inspired strategy for compounding graphene@NiFe₂O₄ flexible films for strong electromagnetic interference shielding and absorption. *Nat. Commun.* **15**, 5486 (2024).
- [R22] Lv, H. *et al.* Staggered circular nanoporous graphene converts electromagnetic waves into electricity. *Nat. Commun.* **14**, 1982 (2023).
- [R23] Geim, A. K. & Novoselov, K. S. The rise of graphene. *Nat. Mater.* **6**, 183–191 (2007).

Original comment 3-2:

There is no explanation why the combination is called "metamaterial".

REPLY:

Thank you very much for this valuable comment. As you know, in a broad sense, the periodic structure that can give electromagnetic materials beyond the properties of materials in nature can be called metamaterials. Now more sub-wavelength (the size is equal to or smaller than the working wavelength) is used to define metamaterials ^[R24-R26].

In this work, we designed the geometric structure of GAF materials based on electromagnetic theory, and transformed the isotropic GAF in plane into anisotropic GAFM in plane by constructing slits with

specific structures, thus endowing it with polarization sensitive properties, which is a characteristic that natural materials do not have. Specifically, the EMI SE is lowest when the incident electric field is oriented parallel to the short side of the slit, while the EMI SE is highest when the incident electric field polarization is orthogonal to the short side of the slit. Therefore, when the angle between the slit orientation (the long side direction) in GAFM and the direction of the electric field is rotated from 0° to 90° , the EMI SE changes from maximum value (almost completely shielding) to minimum value (almost entirely transmitting). Therefore, we believe GAFM is a type of metamaterial.

Similarly, to take advantages of GAFM and CF, we compounded these two artificial materials and further boost the manipulation of EMI SE, including larger polarization difference and fully polarized shielding. When the slit orientation in GAFM and the fiber axis direction of CF is parallel (GAFM//CF), the rotational manipulation range of GAFM/CF is maximized, achieving an enhanced polarization difference and expanding manipulated range. On the other hand, when the slit orientation in GAFM and the fiber axis direction of CF is perpendicular (GAFM \perp CF), the GAFM/CF achieves shortened polarization differences and the **highest full polarization shielding**. This is also a characteristic that natural materials do not have. Hence, we believe GAFM/CF is a type of metamaterial.

We have added relevant descriptions in the revised manuscript (Page 5-6, Line 99-108) and the revised Supporting Information (Page 4-6).

“In order to further boost the manipulation of EMI SE, including larger polarization difference and fully polarized shielding, we compounded GAFM and CF to take advantages of these two artificial materials. When the slit orientation in GAFM and the fiber axis direction of CF is parallel (GAFM//CF), the rotational manipulation range of GAFM/CF is maximized, achieving an enhanced polarization difference and expanding manipulated range. On the other hand, when the slit orientation in GAFM and the fiber axis direction of CF is perpendicular (GAFM \perp CF), the GAFM/CF achieves shortened polarization differences and fully polarized EMI shielding. This is also a characteristic that natural materials do not have. Hence, we believe GAFM/CF is a type of metamaterial.”

[R24] Cheng, Y. Z. *et al.* Design, fabrication and measurement of a broadband polarization-insensitive metamaterial absorber based on lumped elements. *J. Appl. Phys.* **111**, 044902 (2012).

[R25] Tuong, P. V., Park, J. W., Kim, Y. J., Yoo, Y. J. & Lee, Y. P. Broadband Reflection of Polarization Conversion by 90° in Metamaterial. *J. Korean Phys. Soc.* **64**, 1116–1119 (2014).

[R26] Zhang, C. *et al.* Broadband metamaterial for optical transparency and microwave absorption. *Appl. Phys. Lett.* **110**, 143511 (2017).

Author's Responses to Comments

Dear editors and reviewers,

The revised manuscript has been greatly benefited from your constructive comments and valuable suggestions. We would like to thank you sincerely for the time and effort to help improve the presentation and qualification of this manuscript.

Response to Reviewer #2

REPLY:

We deeply appreciate your insightful feedback and valuable advices, which have greatly contributed to the improvement of our manuscript. Your time and effort in enhancing both the clarity and quality of our work are truly appreciated. We are grateful for the precision, patience, and meticulous attention to detail you have shown throughout the review process. Your constructive comments have played a crucial role in refining the manuscript. We hope that the revisions now meet your expectations. In our response, we have carefully addressed each of your comments.

Original comment 2-1:

Where is the role of metamaterials in the design? The periodicity of the structure is 18 mm, while the device frequency is in X-band. Thus, the dimension is not in quarter-wavelength calling for possibility of spurious modes which has not been studied.

REPLY:

Thank you for this enlightening comment.

1. The role of metamaterials in the design can be considered as a frequency-selective surface (FSS) with polarization sensitivity. Specifically, as a bandpass FSS structure with GAFM, it can achieve the lowest EMI shielding efficiency (“ON” state) only when the polarization direction is parallel to the short side, and the EMI shielding efficiency is the highest (“OFF” state) when the polarization direction is perpendicular to the short side. It is worth noting that the period of our unit cell is approximately half the operating wavelength. It is appropriate to refer to the structure as a metamaterial, given its unique electromagnetic properties and subwavelength size—the definition that has also been adopted in

numerous professional publications.^[R1-R6] In fact, the real innovation lies in our **extension of classical electromagnetic equivalent circuit theory—traditionally applied to metallic materials—to advanced carbon materials with microstructures**, such as graphene-assembled films (GAF). This shift in theoretical approach allows us to not only **elucidate the EMI shielding mechanism and polarization manipulation mechanism** of advanced carbon materials, but also to develop serialized EMI SE manipulation metamaterials with some record-breaking performances—specifically, the **widest polarization difference manipulation range (enhanced polarization sensitivity)**, the **highest full polarization shielding (full polarization shielding)**, and the **most significant polarization switching effect (switching efficiency for “ON” and “OFF” states)**, simultaneously. This is a major innovation differing from previous work on EMI SE manipulation with carbon materials, which performance is far inferior to our work.

2. As you pointed out, the period of our metamaterial structure is 18 mm, which is close to half the wavelength at the operating frequency, rather than a quarter wavelength. We believe your comment is highly insightful and effectively highlights a limitation in our explanation of the frequency selection mechanism.

Figure R1. Simulated S-parameters in Y-polarization direction.

The principle can be demonstrated by analyzing the electric field and surface distribution of the metamaterial. As an example, in the Y-polarization direction, the S-parameters exhibit two clear resonance points, as illustrated in **Figure R1**. Due to the Y-polarized incident electromagnetic wave, the electric potentials on the two long sides of the metamaterial are different. **Figure R2(a)** illustrates that the electric field on the metamaterial at 10 GHz induced by the Y-polarized wave exhibits a

symmetric distribution with respect to the Y-axis. In the condition where the structure length equals half the wavelength, the electric field antinodes coincide. According to the surface current distribution shown in the **Figure R2(b)**, when the current path is an odd multiple of half the wavelength, the antinodes of the current wave coincide, leading to maximum current amplitude. At this point, both the voltage and current amplitudes reach their maximum values, indicating that the system is in a resonant state. **Therefore, it is shown that the structure does produce 1/2 wavelength resonance under the excitation of Y-polarized wave.**

If the length of the structure corresponds to the quarter wavelength, this spatial wave incidence is different from the resonance of the traveling wave structure (such as the 1/4 resonator of the microstrip). The electric field and current amplitudes cannot achieve antinode superposition, as illustrated in **Figure R3(a)** and **R3(b)**. As a result, resonance cannot occur, making it impossible to achieve frequency-selective behavior. In other words, **the resonant frequencies of our metamaterial occur at odd multiples of half wavelength, while no resonance is observed at the quarter-wavelength condition (5 GHz)**, which can be demonstrated by reflection coefficient shown in **Figure R1(a)**.

To further verify the validity of our design principle, we simulated the electric field and current distributions of the metamaterial at a higher-order resonant frequency (25 GHz), as illustrated in **Figure R4(a)** and **R4(b)**. At this frequency, the length of the metamaterial corresponds to 1.5 wavelengths (odd multiples of half wavelength), which is consistent with our design concept.

To demonstrate the polarization sensitivity of the designed metamaterial, we simulated the resonant frequencies in different polarization directions, as shown in **Figure R5(a)**. For the X-polarization direction, the resonance still occurs when the length of the short side of the metamaterial corresponds to half wavelength, and the antinodes are superimposed as displayed in **Figure R5(b)**. There was no quarter-wavelength resonance at low frequencies either.

The difference in geometric dimensions along the X and Y axes imparts polarization sensitivity to the metamaterial, resulting in clearly distinguishable resonant frequencies in different polarization across a frequency range.

Figure R2. (a) Electric field distribution and (b) surface current distribution at 10 GHz.

Figure R3. (a) Electric field distribution and (b) surface current distribution at 5 GHz.

Figure R4. (a) Electric field distribution and (b) surface current distribution at 25 GHz.

Figure R5. (a) Simulated S-parameters in X-polarization direction. (b) Surface current distribution at 40 GHz.

Finally, we sincerely thank you for your comment, which has helped improve the completeness of our explanation on metamaterials for frequency selection with high polarization sensitivity, enhancing the overall rigor of the paper. According to your suggestion, we added the explanation of the frequency selection mechanism to the revised manuscript (Page 8, Lines 7-13):

“This occurs because when the length of the periodic metamaterial structure equals half of the corresponding wavelength, the wave antinodes align spatially, causing both the voltage and current amplitudes to reach their maximum values, which signifies that the system has entered a resonant state (passband). For different polarizations, the side lengths in the X and Y directions are different, which results in different resonant frequencies, thereby obtaining a designable high polarization sensitivity at the same operating frequency (See Supplementary Note 2).”

In addition, the design principle has been included in (Page 35-38) the revised Supplementary Information:

“Supplementary Note 2. The principle of high-polarization sensitivity of GAFM:

The principle can be demonstrated by analyzing the electric field and surface distribution of the metamaterial. As an example, in the Y-polarization direction, the S-parameters exhibit two clear resonance points, as illustrated in **Supplementary Fig. 17**. Due to the Y-polarized incident electromagnetic wave, the electric potentials on the two long sides of the metamaterial are different. **Supplementary Fig. 18(a)** illustrates that the electric field on the metamaterial at 10 GHz induced by

the Y-polarized wave exhibits a symmetric distribution with respect to the Y-axis. In the condition where the structure length equals half the wavelength, the electric field antinodes coincide. According to the surface current distribution shown in the **Supplementary Fig. 18(b)**, when the current path is an odd multiple of half the wavelength, the antinodes of the current wave coincide, leading to maximum current amplitude. At this point, both the voltage and current amplitudes reach their maximum values, indicating that the system is in a resonant state. Therefore, it is shown that the structure does produce $1/2$ wavelength resonance under the excitation of Y-polarized wave.

If the length of the structure corresponds to the quarter wavelength, this spatial wave incidence is different from the resonance of the traveling wave structure (such as the $1/4$ resonator of the microstrip). The electric field and current amplitudes cannot achieve antinode superposition, as illustrated in **Supplementary Fig. 19(a)** and **19(b)**. As a result, resonance cannot occur, making it impossible to achieve frequency-selective behavior. In other words, the resonant frequencies of our metamaterial occur at odd multiples of half wavelength, while no resonance is observed at the quarter-wavelength condition (5 GHz), which can be demonstrated by reflection coefficient shown in **Supplementary Fig. 17(a)**.

To further verify the validity of our design principle, we simulated the electric field and current distributions of the metamaterial at a higher-order resonant frequency (25 GHz), as illustrated in **Supplementary Fig. 20(a)** and **20(b)**. At this frequency, the length of the metamaterial corresponds to 1.5 wavelengths (odd multiples of half wavelength), which is consistent with our design concept.

To demonstrate the polarization sensitivity of the designed metamaterial, we simulated the resonant frequencies in different polarization directions, as shown in **Supplementary Fig. 21(a)**. For the X-polarization direction, the resonance still occurs when the length of the short side of the metamaterial corresponds to half wavelength, and the antinodes are superimposed as displayed in **Supplementary Fig. 21(b)**. There was no quarter-wavelength resonance at low frequencies either.

The difference in geometric dimensions along the X and Y axes imparts polarization sensitivity to the metamaterial, resulting in clearly distinguishable resonant frequencies in different polarization across a frequency range.

Supplementary Fig. 17. Simulated S-parameters in Y-polarization direction.

Supplementary Fig. 18. (a) Electric field distribution and (b) surface current distribution at 10 GHz.

Supplementary Fig. 19. (a) Electric field distribution and (b) surface current distribution at 5 GHz.

Supplementary Fig. 20. (a) Electric field distribution and (b) surface current distribution at 25 GHz.

Supplementary Fig. 21. (a) Simulated S-parameters in X-polarization direction. (b) Surface current distribution at 40 GHz.”

- [R1] Zhou, Y., *et al.* Ultra-broadband metamaterial absorbers from long to very long infrared regime. *Light Sci. Appl.* **10**, 138 (2021).
- [R2] Huang, Y., *et al.* Topological transition of Pancharatnam-Berry phase in a nonlocal twisted bilayer metasurface. *Sci. Rep.* **15**, 11182 (2025).
- [R3] Liu, S., *et al.* Anomalous refraction and nondiffractive bessel-beam generation of Terahertz waves through transmission-type coding metasurfaces. *ACS Photonics* **3**, 1968-1977, (2016).
- [R4] Tian, H. W., *et al.* Solar-powered light-modulated microwave programmable metasurface for sustainable wireless communications. *Nat. Commun.* **16**, 2524 (2025).
- [R5] Huang, L., *et al.* Three-dimensional optical holography using a plasmonic metasurface. *Nat. Commun.* **4**, 2808 (2013).

[R6] Wang, Y., *et al.* Vanadium dioxide enabled polarization insensitive tunable broadband terahertz metamaterial absorber. *Sci. Rep.* **15**, 10140 (2025).

Original comment 2-2:

The EMI measurements should be carried out step by step and shown accordingly.

REPLY:

Thank you for your valuable advice. Electromagnetic interference (EMI) shielding effectiveness (SE) in this work is measured using the radiated field method, which is a standard technique commonly employed for metasurface reflection or transmission characterization.^[R1–R6] Taking GAFM as an example, the specific steps of its EMI SE measurement are as follows:

1 Set up the test environment

In the radiated field method, the transmission characteristics of the sample are measured by placing it between the transmitting and receiving antennas. The experimental setups for EMI SE measurement of the proposed GAFM are illustrated in **Figure R6**. To ensure that the plane wave is normally incident on the metamaterial, the measurement must satisfy the far-field condition—specifically, the sample under measurement should be located within the far-field region of both the transmitting and receiving antennas. The far-field calculation conditions are:^[R7]

$$R \geq \frac{2D^2}{\lambda} \quad (1)$$

Where R is the minimum distance between the antenna and the sample, D is the diameter of the antenna aperture, and λ is the operating wavelength. The distance between the reflector and the two horn antennas is 0.3 m. A 216 mm × 216 mm rectangular hole is made in the absorber, into which the sample is inserted for EMI SE measurement. This configuration helps reduce diffraction propagation. To ensure the accuracy of the experiment, the entire measurement in this work was carried out in the anechoic chamber.

Figure R6. Experimental setup for EMI SE measurement calibration.

2 Calibrate the system and apply time gating

The transmission coefficients through air were measured without a sample in place as a reference baseline. In addition, time gating is applied by converting the frequency-domain S-parameters into the time domain via a Fast Fourier transform (FFT). In the time domain, the signal directly associated with the sample along the main propagation path (the first transmission wave) can be identified and isolated. Stray signals in other delay intervals are effectively suppressed. The isolated signal is then transformed back into the frequency domain using an Inverse Fourier transform (IFT) to obtain purified transmission coefficient data. This process eliminates the influence of stray reflections, diffraction, and other disturbances on the measurement results. **Figure R7** illustrates the configuration of the time gating setup. We use the transmission coefficient obtained after applying time gating, as the reference (full transmission) for subsequent measurements.

Figure R7. Configuration of the time gating window.

3 Obtain transmission coefficient

Place the metamaterial in the far field between the two horn antennas, with its shorter side aligned parallel to the polarization direction of the horns, as displayed in **Figure R8**. The EM wave is transmitted by horn antenna 1 and passed through the sample to be received by horn antenna 2. Through this process, the transmission coefficient of the sample is obtained.

Figure R8. Experimental setup for testing GAFM in Y-polarization direction.

4 Rotate different angles and measure EMI characteristics

As shown in **Figure R9**, we rotate the two horn antennas to different angles and measure the EMI characteristics following the same steps. The measured original transmission coefficient (S_{21}) and the corresponding EMI SE result distribution are shown in **Figure R10(a)** and **R10(b)**. The S_{21} parameter reflects the signal attenuation through the structure. A smaller S_{21} value indicates diminished signal penetration, corresponding to enhanced EMI SE.

Figure R9. Experimental setup for metamaterial measurements under various polarization angles: (a) 30°, (b) 45°, (c) 60°, and (d) 90° polarization.

Figure R10. (a) The measured transmission coefficient (S_{21}). (b) The corresponding EMI SE result.

In addition, the waveguide method is also commonly used for EMI SE measurement; **however, its application is somewhat limited by the size of the waveguide port, making it less versatile than the radiated field method.** In this work, to ensure the accuracy and rigor of the measurements, we also employ the waveguide method for relevant measurements. The specific steps are as follows:

1. Instrument Calibration

Figure R11. Calibration of the Vector Network Analyzer. (a) Open the calibration interface. The waveguide adapter of port 1 is sequentially connected to (b) a short circuit, (c) a quarter-wave resonator and the short circuit, and (d) a waveguide load for calibration. The waveguide adapter of port 2 is sequentially connected to (e) a short circuit, (f) a quarter-wave resonator and the short circuit, and (g) a waveguide load for calibration. (h) Connect the waveguide adapters of ports 1 and 2 to each other for calibration. (i) Calibration interface that calibration has been completed.

As shown in **Figure R11**, the calibration of the vector network analyzer is performed first.

- (1) Connect the waveguide adapter of port 1 (left side) to a short circuit, then select the “Short” calibration for port 1.
- (2) Connect the waveguide adapter of port 1 to a quarter-wave resonator and the short circuit, then select the “Offset short” calibration for port 1.
- (3) Connect the waveguide adapter of port 1 to a waveguide load, then select the “Load” calibration for port 1.
- (4) Connect the waveguide adapter of port 2 (right side) to a short circuit, then select the “Short” calibration for port 2.
- (5) Connect the waveguide adapter of port 2 to a quarter-wave resonator and the short circuit, then select the “Offset short” calibration for port 2.
- (6) Connect the waveguide adapter of port 2 to a waveguide load, then select the “Load” calibration for port 2.

- (7) Finally, connect the waveguide adapters of ports 1 and 2 to each other, then select the “Thru” calibration to complete the entire calibration procedure.

2. EMI SE Testing Procedure of GAFM

Figure R12. (a) The center of the GAFM is aligned with the center of the quarter-wave resonator, and both are fixed into the waveguide adapter. The following vector network analyzer display results during the EMI SE test at various rotational angles: (b) 0° , (c) 30° , (d) 45° , (e) 60° , (f) 90° .

- (1) After completing the calibration, first align the center of the GAFM with the center of the quarter-wave resonator, then fix both the GAFM and the quarter-wave resonator into the waveguide adapter, as shown in **Figure R12(a)**. This position is considered 0° , and EMI SE test data is recorded.
- (2) Keep the center of the GAFM aligned with the center of the quarter-wave resonator, then rotate the sample clockwise to 0° . Record the EMI SE test data as **Figure R12(b)**.
- (3) Keep the center of the GAFM aligned with the center of the quarter-wave resonator, then rotate the sample clockwise to 30° . Record the EMI SE test data as **Figure R12(c)**.
- (4) Keep the center of the GAFM aligned with the center of the quarter-wave resonator, then rotate the sample clockwise to 45° . Record the EMI SE test data as **Figure R12(d)**.
- (5) Keep the center of the GAFM aligned with the center of the quarter-wave resonator, then rotate the sample clockwise to 60° . Record the EMI SE test data as **Figure R12(e)**.
- (6) Keep the center of the GAFM aligned with the center of the quarter-wave resonator, then rotate the sample clockwise to 90° . Record the EMI SE test data as **Figure R12(f)**.

Figure R13. Schematic of EMI SE testing of GAFM using the waveguide method at different rotational angles. The testing procedure includes the following angles: (a) 0°, (b) 30°, (c) 45°, (d) 60°, (e) 90°, and (f) a schematic showing the connection between the quarter-wave resonator and the waveguide adapter during the testing process.

The rotation process is shown in **Figure R13(a-e)**. First, align the center of the GAFM with the center of the quarter-wave resonator. Then, sequentially rotate the sample clockwise to different angles, including 0°, 30°, 45°, 60°, and 90°, and fix both the GAFM and the quarter-wave resonator into the waveguide adapter as shown in **Figure R13(f)**. Record the corresponding EMI SE test data (**Figure R14**). Throughout the entire testing process, ensure that the center of the GAFM remains aligned with the center of the quarter-wave resonator.

Figure R14. EMI SE of GAFM using the waveguide method at different rotational angles: 0°, 30°, 45°, 60° and 90°.

We apologize for not describing the measurement steps in detail. Thanks to your valuable suggestions, we have added the detailed EMI SE measurement steps in the revised Supplementary Information (Page 34, Lines 13-18) as follows:

“We performed EMI SE measurements using the radiated field method,^[S40-S41] placing the GAFM in the far-field region between two horn antennas. The EMI characteristics were evaluated by measuring the transmission coefficient between the two antennas, constrained by time gating. By changing the polarization direction of the horn antennas, we measured the response of the GAFM to electromagnetic waves with different polarizations.

[S40] Wang, J. & Yang, R. Generating High-Purity Directive Circularly Polarized Beams From Conformal Anisotropic Holographic Metasurfaces. *IEEE Transactions on Antennas and Propagation* 70, 10718-10723 (2022).

[S41] International Electrotechnical Commission. Specification for radio disturbance and immunity measuring apparatus and methods – Part 2-3: Methods of measurement of disturbances and immunity – Radiated disturbance measurements. IEC CISPR 16-2-3, Ed. 4.0 (2016)”

In addition, we have included a detailed description of the EMI SE measurement procedures in the revised Supplementary Information (Page 39-43):

“Supplementary Note 3. The EMI SE measurement procedures:

Electromagnetic interference (EMI) shielding effectiveness (SE) is measured using the radiated field method, which is a standard technique commonly employed for metasurface reflection or transmission characterization. Taking GAFM as an example, the specific steps of its EMI SE measurement are as follows:

1 Set up the test environment

In the radiated field method, the transmission characteristics of the sample are measured by placing it between the transmitting and receiving antennas. The experimental setups for EMI SE measurement of the proposed GAFM are illustrated in **Supplementary Fig. 22**. To ensure that the plane wave is normally incident on the metamaterial, the measurement must satisfy the far-field condition—specifically, the sample under measurement should be located within the far-field region of both the transmitting and receiving antennas. The far-field calculation conditions are:

$$R \geq \frac{2D^2}{\lambda} \quad (1)$$

Where R is the minimum distance between the antenna and the sample, D is the diameter of the antenna aperture, and λ is the operating wavelength. The distance between the reflector and the two horn antennas is 0.3 m. A 216 mm \times 216 mm rectangular hole is made in the absorber, into which the sample is inserted for EMI SE measurement. This configuration helps reduce diffraction propagation. To ensure the accuracy of the experiment, the entire measurement in this work was carried out in the anechoic chamber.

Supplementary Fig. 22. Experimental setup for EMI SE measurement calibration.

2 Calibrate the system and apply time gating

The transmission coefficients through air were measured without a sample in place as a reference baseline. In addition, time gating is applied by converting the frequency-domain S -parameters into the time domain via a Fast Fourier Transform (FFT). In the time domain, the signal directly associated with the sample along the main propagation path (the first transmission wave) can be identified and isolated. Stray signals in other delay intervals are effectively suppressed. The isolated signal is then transformed back into the frequency domain using an Inverse Fourier Transform (IFT) to obtain purified transmission coefficient data. This process eliminates the influence of stray reflections, diffraction, and other disturbances on the measurement results. The **Supplementary Fig. 23** illustrates the configuration of the time gating setup. We use the transmission coefficient obtained after applying time gating, as the reference (full transmission) for subsequent measurements.

Supplementary Fig. 23. Configuration of the time gating window.

3 Obtain transmission coefficient

Place the metamaterial in the far field between the two horn antennas, with its shorter side aligned parallel to the polarization direction of the horns, as displayed in **Supplementary Fig. 24**. The EM wave is transmitted by horn antenna 1 and passed through the sample to be received by horn antenna 2. Through this process, the transmission coefficient of the sample is obtained.

Supplementary Fig. 24. Experimental setup for testing GAFM in Y-polarization direction.

4 Rotate different angles and measure EMI characteristics

As shown in **Supplementary Fig. 25**, we rotate the two horn antennas to different angles and measure the EMI characteristics following the same steps. The measured original transmission coefficient (S_{21}) and the corresponding EMI SE result distribution are shown in **Supplementary Fig. 26(a)** and **26(b)**.

The S_{21} parameter reflects the signal attenuation through the structure. A smaller S_{21} value indicates diminished signal penetration, corresponding to enhanced EMI SE.

Supplementary Fig. 25. Experimental setup for metamaterial measurements under various polarization angles: (a) 30°, (b) 45°, (c) 60°, and (d) 90° polarization.

Supplementary Fig. 26. (a) The measured transmission coefficient (S_{21}). (b) The corresponding EMI SE result.”

References:

[R1] Li, F. *et al.* Flexible intelligent microwave metasurface with shape-guided adaptive programming. *Nat. Commun.* **16**, 3161 (2025).

- [R2] Li, W., Yu, Q., Qiu, J. H. & Qi, J. Intelligent wireless power transfer via a 2-bit compact reconfigurable transmissive-metasurface-based router. *Nat. Commun.* **15**, 2807 (2024).
- [R3] Wang, J. & Yang, R. Generating High-Purity Directive Circularly Polarized Beams From Conformal Anisotropic Holographic Metasurfaces. *IEEE Transactions on Antennas and Propagation* **70**, 10718-10723 (2022).
- [R4] Wang, X. *et al.* High-performance cost efficient simultaneous wireless information and power transfers deploying jointly modulated amplifying programmable metasurface. *Nat. Commun.* **14**, 6002, (2023).
- [R5] Wang, Y. *et al.* Dual-Band Metasurface With Extreme Angular-Asymmetric Transmission and Frequency Selection Based on Resonant Coupling Effect. *IEEE Transactions on Antennas and Propagation* **71**, 7656-7660 (2023).
- [R6] International Electrotechnical Commission. Specification for radio disturbance and immunity measuring apparatus and methods – Part 2-3: Methods of measurement of disturbances and immunity – Radiated disturbance measurements. **IEC CISPR 16-2-3**, Ed. 4.0 (2016)
- [R7] IEEE. IEEE Standard Test Procedures for Antennas. **IEEE Std 149–1979** (Institute of Electrical and Electronics Engineers, 1979).

Original comment 2-3:

The equivalent circuit design mentioned in Fig. 3 is not correct as the wave impinges on the structure, it undergoes transmission along the substrate; thus, there needs a consideration of an equivalent transmission line. See <https://doi.org/10.1364/AO.431821> for details. The circuit model response and the simulated/ measured results of the S-parameters must be compared.

REPLY:

Thank you for your valuable comments. We have read this paper carefully. Since the EM wave impinges on the structure, electromagnetic waves do propagate on the substrate (mainly in the form of the electric field) and resonate on the conductor (mainly in the form of current) as shown in **Figure R15(a)** and **R15(b)**.

Figure R15. (a) Electric field distribution in the cross-section of the metamaterial and (b) corresponding surface current distribution on the metamaterial under y polarization.

Figure R16. (a) Schematic diagram of equivalent circuit model and (b) related parameters of the GAFM.

Unlike the structures reported in the references, GAFM consists solely of a conductive layer. In other words, it can be understood that the air on both sides of the GAFM effectively serves as its substrate. Therefore, as shown in **Figure R15(a)** and consistent with your observation, electromagnetic waves propagate through the air on both sides as if along a substrate, and the transmission line effect must be taken into account. This corresponds to Part 1 in the equivalent circuit model plotted in **Figure R16**. As shown in **Figure R15(b)**, resonance occurs within the GAFM conductive structure, and the associated RLC series-parallel configuration and resonance effects must be considered. This corresponds to Part 2 in the equivalent circuit model. Therefore, the constructed equivalent circuit takes into account both the

transmission line effect of the substrate and the resonance behavior of the conductive structure, as shown above (**Figure R16**). According to transmission line theory, the transfer matrix of this structure can be expressed as:

$$\begin{bmatrix} A & B \\ C & D \end{bmatrix} = \begin{bmatrix} \cos \theta & jZ_0 \sin \theta \\ j\frac{1}{Z_0} \sin \theta & \cos \theta \end{bmatrix} \cdot \begin{bmatrix} 1 & 0 \\ j\omega C & 1 \end{bmatrix} \cdot \begin{bmatrix} 1 & 0 \\ \frac{1}{R+j\omega L} & 1 \end{bmatrix} \cdot \begin{bmatrix} \cos \theta & jZ_0 \sin \theta \\ j\frac{1}{Z_0} \sin \theta & \cos \theta \end{bmatrix} = \begin{bmatrix} -1 & -jZ_0^2 \omega C - \frac{Z_0^2}{R+j\omega L} \\ 0 & -1 \end{bmatrix} \quad (2)$$

Where $\theta = \pi/2$, $Z_0 = 377 \Omega$.

Its scattering matrix can be expressed as:

$$\begin{bmatrix} S_{11} & S_{12} \\ S_{21} & S_{22} \end{bmatrix} = \begin{bmatrix} \frac{A+B/Z_0-CZ_0-D}{A+B/Z_0+CZ_0+D} & \frac{2(AD-BC)}{A+B/Z_0+CZ_0+D} \\ \frac{2}{A+B/Z_0+CZ_0+D} & \frac{-A+B/Z_0-CZ_0+D}{A+B/Z_0+CZ_0+D} \end{bmatrix} \quad (3)$$

Its impedance matrix can be expressed as:

$$\begin{bmatrix} Z_{11} & Z_{12} \\ Z_{21} & Z_{22} \end{bmatrix} = \begin{bmatrix} \frac{A}{C} & \frac{AD-BC}{C} \\ \frac{1}{C} & \frac{D}{C} \end{bmatrix} \quad (4)$$

The input impedance can be expressed as:

$$Z_{in} = \frac{AZ_0 + B}{CZ_0 + D} \quad (5)$$

As plotted in Figure 3e and Figure 3f, the simulated transmission coefficient of the mathematical model, equivalent circuit model and full wave model are consistent for both Y polarization direction and X polarization direction, verifying that the design of GAFM is feasible from theory to practice. To prevent any misunderstanding by readers regarding our consideration of the substrate's transmission line effect and explain the theory more clearly, we have added the following clarification in the revised manuscript (Page 10, Lines 6-19):

“Then, we modeled the GAFM as a combination of capacitance-inductance-resistance and calculated parameters of GAFM based on equivalent circuit model (Fig. 3d).” has been changed to “**Then, we modeled the GAFM as a combination of capacitance-inductance-resistance and calculated parameters of GAFM based on the equivalent circuit model (Fig. 3d).** It is worth noting that the GAFM is a pure

conductive film without any substrate. Therefore, in this work, the equivalent circuit model is established using air as the substrate for the corresponding transmission line.³³

According to transmission line theory, the transfer matrix of this structure can be expressed as:

$$\begin{bmatrix} A & B \\ C & D \end{bmatrix} = \begin{bmatrix} \cos \theta & jZ_0 \sin \theta \\ j\frac{1}{Z_0} \sin \theta & \cos \theta \end{bmatrix} \cdot \begin{bmatrix} 1 & 0 \\ j\omega C & 1 \end{bmatrix} \cdot \begin{bmatrix} 1 & 0 \\ \frac{1}{R+j\omega L} & 1 \end{bmatrix} \cdot \begin{bmatrix} \cos \theta & jZ_0 \sin \theta \\ j\frac{1}{Z_0} \sin \theta & \cos \theta \end{bmatrix} = \begin{bmatrix} -1 & -jZ_0^2\omega C - \frac{Z_0^2}{R+j\omega L} \\ 0 & -1 \end{bmatrix} \quad (1)$$

Where $\theta = \pi/2$, $Z_0 = 377 \Omega$.

Figure 3. Verification of parametric quantization and feasibility of transitioning from the microscopic structural model to the macroscopic electromagnetic equivalent circuit model. The

SEM cross-sectional image of (a₁) GAF and (a₂) Cu foil. Electric potential distribution of GAF when applying an electric field in the (b₁) in-plane direction and (c₁) thickness direction; electric potential distribution of Cu foil when applying an electric field in the (b₂) in-plane direction and (c₂) thickness direction. (d) Schematic diagram of equivalent circuit model and related parameters of the GAFM. Simulated transmission coefficient of the mathematical model (calculation value, named “CAL”), equivalent circuit model (Advanced Design System, named “ADS”) and full wave model (Computer Simulation Technology, named “CST”) for (e) Y polarization direction and (f) X polarization direction. Field map of electric field distribution of GAFM for (g) Y-polarized and (h) X-polarized electromagnetic waves at 10 GHz. Field map of surface current distribution of GAFM for (i) Y-polarized and (j) X-polarized electromagnetic waves at 10 GHz. Dependence of ΔEMI SE in (k) Y polarization and (l) X polarizations on the lengths of *sl* at 7-13 GHz (*sw* = 15.0 mm). Dependence of ΔEMI SE in (m) Y polarization and (n) X polarizations on the lengths of *sw* at 7-13 GHz (*sl* = 3.5 mm).

Its scattering matrix can be expressed as:

$$\begin{bmatrix} S_{11} & S_{12} \\ S_{21} & S_{22} \end{bmatrix} = \begin{bmatrix} \frac{A+B/Z_0-CZ_0-D}{A+B/Z_0+CZ_0+D} & \frac{2(AD-BC)}{A+B/Z_0+CZ_0+D} \\ \frac{2}{A+B/Z_0+CZ_0+D} & \frac{-A+B/Z_0-CZ_0+D}{A+B/Z_0+CZ_0+D} \end{bmatrix} \quad (2)$$

Its impedance matrix can be expressed as:

$$\begin{bmatrix} Z_{11} & Z_{12} \\ Z_{21} & Z_{22} \end{bmatrix} = \begin{bmatrix} \frac{A}{C} & \frac{AD-BC}{C} \\ \frac{1}{C} & \frac{D}{C} \end{bmatrix} \quad (3)$$

The input impedance can be expressed as:

$$Z_{in} = \frac{AZ_0 + B}{CZ_0 + D} \quad (4)$$

Reference:

[33] Paul, A., Nilotpal, Bhattacharyya, S. & Dwivedi, S. Design and mathematical analysis of a metasurface-based THz bandpass filter with an equivalent circuit model. *Appl. Opt.* **60**, 6429-6437 (2021).”

In fact, the GAFM initially prepared was attached to an 80-micron PET substrate during the early stages of fabrication. In this work, to ensure the rigor and accuracy of the GAFM-related analysis and measurements, the PET layer was removed prior to measurement. Since the dielectric constant of PET is

relatively low ($\epsilon_r = 3.3$), the presence of a PET substrate has a minimal impact on the EMI response and its control. Furthermore, when incorporating the PET substrate into the equivalent circuit model, the simulation and calculation results remain highly consistent.

Based on your suggestions, we added an equivalent transmission line of dielectric substrate to the circuit. The final equivalent circuit model is displayed in **Figure R17**.

Figure R17. Schematic diagram of equivalent circuit model with dielectric substrate.

According to transmission line theory, the transfer matrix of this structure can be expressed as:

$$\begin{aligned} \begin{bmatrix} A & B \\ C & D \end{bmatrix} &= \begin{bmatrix} \cos \theta & jZ_0 \sin \theta \\ j\frac{1}{Z_0} \sin \theta & \cos \theta \end{bmatrix} \cdot \begin{bmatrix} 1 & 0 \\ j\omega C & 1 \end{bmatrix} \cdot \begin{bmatrix} 1 & 0 \\ \frac{1}{R+j\omega L} & 1 \end{bmatrix} \cdot \begin{bmatrix} \cos \beta t & jZ_s \sin \beta t \\ j\frac{1}{Z_s} \sin \beta t & \cos \beta t \end{bmatrix} \cdot \begin{bmatrix} \cos \theta & jZ_0 \sin \theta \\ j\frac{1}{Z_0} \sin \theta & \cos \theta \end{bmatrix} \\ &= \begin{bmatrix} \frac{j^2 [jZ_s (Cj\omega(R+j\omega L)+1)\sin \beta t + (R+j\omega L)\cos \beta t]}{R+j\omega L} & \frac{j^2 Z_0^2 [Z_s (Cj\omega(R+j\omega L)+1)\cos \beta t + j(R+j\omega L)\sin \beta t]}{Z_s (R+j\omega L)} \\ \frac{j^3 Z_s \sin \beta t}{Z_0^2} & j^2 \cos \beta t \end{bmatrix} \quad (6) \end{aligned}$$

Where $\theta = \pi/2$, $Z_0 = 377 \Omega$, $Z_s = 207 \Omega$, $t = 80 \mu\text{m}$, $\beta = \omega\sqrt{\mu\epsilon}$.

Its scattering matrix can be expressed as:

$$\begin{bmatrix} S_{11} & S_{12} \\ S_{21} & S_{22} \end{bmatrix} = \begin{bmatrix} \frac{A+B/Z_0-CZ_0-D}{A+B/Z_0+CZ_0+D} & \frac{2(AD-BC)}{A+B/Z_0+CZ_0+D} \\ \frac{2}{A+B/Z_0+CZ_0+D} & \frac{-A+B/Z_0-CZ_0+D}{A+B/Z_0+CZ_0+D} \end{bmatrix} \quad (7)$$

Its impedance matrix can be expressed as:

$$\begin{bmatrix} Z_{11} & Z_{12} \\ Z_{21} & Z_{22} \end{bmatrix} = \begin{bmatrix} \frac{A}{C} & \frac{AD-BC}{C} \\ \frac{1}{C} & \frac{D}{C} \end{bmatrix} \quad (8)$$

The input impedance can be expressed as:

$$Z_{in} = \frac{AZ_0 + B}{CZ_0 + D} \quad (9)$$

As per your request, and to verify the accuracy of the proposed equivalent circuit model with respect to the design parameters of the frequency-selective surface, we evaluated the circuit response using three different methods: (1) analytical calculation based on a two-port microwave network model (abbreviated as "CAL"), (2) circuit simulation using AWR software, and (3) full-wave electromagnetic simulation using CST Microwave Studio. The circuit element parameters in the Y polarization direction are as follows: $R = 0.36 \Omega$, $L = 1.588 \text{ nH}$, $C = 0.154 \text{ pF}$. The circuit element parameters in the X polarization direction are as follows: $R = 0.03 \Omega$, $L = 0.158 \text{ nH}$, $C = 0.0366 \text{ pF}$. As shown in **Figure R18**, the simulated transmission coefficients obtained from the mathematical model, the equivalent circuit model, and the full-wave simulation are consistent under both Y- and X-polarization. This consistency confirms the feasibility of the GAFM design from theoretical modeling to practical implementation.

Figure R18. Simulated transmission coefficient of the mathematical model (calculation value, named "CAL"), equivalent circuit model with dielectric substrate (AWR Design Environment, named "AWR") and full wave model (Computer Simulation Technology, named "CST") for (a) Y polarization direction and (b) X polarization direction.

Original comment 2-4:

A comparison table highlighting the novelty of the design compared to the existing one should be mentioned.

REPLY:

Thanks for providing valuable insights that will enhance our work. The key novelty of this work lies in providing a significant conceptual advance by extending classical electromagnetic equivalent circuit theory, traditionally applied to metallic materials, to advanced carbon materials with microstructures (such as GAF). We verified the feasibility of transitioning from the microscopic structural model of carbon materials to the macroscopic electromagnetic equivalent circuit model, which enabled the manipulative design of polarization shielding in carbon-based materials. This is a major departure from previous work on EMI SE manipulation with carbon materials, which has lacked a strong theoretical foundation and has often resulted in lower performance (**Figure R19**, reported in the references from [S4] to [S42]). Furthermore, we have developed serialized EMI SE manipulation metamaterials with a record-breaking performance [including the widest polarization difference manipulation range (enhanced polarization sensitivity), the highest full polarization shielding (full polarization shielding), and the most significant polarization switching effect (switching efficiency)] simultaneously based on advanced carbon materials through design.

Figure R19. Comparison of EMI shielding efficiency for “On” state and “Off” state, and switching efficiency using different strategies (shape change, temperature change, humidity change, electrochemical potential change, and rotation angle change) reported in the references from [S1] to [S39] at X band (8.2 - 12.4 GHz).

Therefore, we have supplemented a comparison table highlighting the novelty of the design compared to the existing ones (**Supplementary Table 1**) in the revised Supplementary Information (Pages 9-15) as follows:

“Supplementary Table 1. Comparison of EMI shielding efficiency for “On” state and “Off” state, switching efficiency using different strategies, and other performances reported in the references at X band (8.2 - 12.4 GHz).

No.	Ref.	Materials	EMI shielding efficiency for “ON” state (η_{ON} , %)	EMI shielding efficiency for “OFF” state (η_{OFF} , %)	Switching efficiency ($\eta_{OFF}-\eta_{ON}$, %)	Full polarization shielding	Enhanced polarization sensitivity	Type
1	S1	Wood-derived carbon/XC-72 NP aerogel	29.2054216 (Y-polarization)	99.7181617 (Y-polarization)	70.5127401	NO	NO	Shape change (compression)
2	S2	TPI-MXene/carbon foam	96.8377223 (Y-polarization)	99.6837722 (Y-polarization)	2.8460499	NO	NO	Shape change (compression)
3	S3	EVA@PPy@Ag foam	76.1768053 (Y-polarization)	99.9999308 (Y-polarization)	23.8231255	NO	NO	Shape change (compression)
4	S4	MF@MXene/Ag NW sponges/PEG	94.2456006 (Y-polarization)	99.9108749 (Y-polarization)	5.6652743	NO	NO	Shape change (compression)
5	S5	WF/CNTs foam	95.9261972 (Y-polarization)	99.9902276 (Y-polarization)	4.0640304	NO	NO	Shape change (compression)
6	S6	PU/CNTs/TPI	99.5831306 (Y-polarization)	99.9668869 (Y-polarization)	0.3837563	NO	NO	Shape change (compression)
7	S7	MF@Ag/PDA/CNTs/waterborne PU	99.6980048 (Y-polarization)	99.9754529 (Y-polarization)	0.2774481	NO	NO	Shape change (compression)
8	S8	Identical patterned anisotropic magnetic liquid metal/PDMS	87.4107459 (Y-polarization)	99.9936904 (Y-polarization)	12.5829445	NO	NO	Shape change (compression)
9	S9	PU/RGO foam-5%	96.8377223 (Y-polarization)	99.4988128 (Y-polarization)	2.6610904	NO	NO	Shape change (compression)

10	S9	PU/RGO foam-10%	99.3690427 (Y- polarization)	99.9936904 (Y- polarization)	0.6246478	NO	NO	Shape change (compression)
11	S10	Pre-linked Ni chains elastomer	36.9042656 (Y- polarization)	99.3690427 (Y- polarization)	62.4647771	NO	NO	Shape change (stretching)
12	S11	CNT@liquid metal/ polyacrylamide/gelatin	49.8812766 (Y- polarization)	99.9900000 (Y- polarization)	50.1087234	NO	NO	Shape change (stretching)
13	S12	Fe/liquid metal/PDMS	99.1290364 (Y- polarization)	99.9999991 (Y- polarization)	0.8709627	NO	NO	Shape change (stretching)
14	S13	3D liquid metal network	99.9645187 (Y- polarization)	99.9999998 (Y- polarization)	0.0354811	NO	NO	Shape change (stretching)
15	S14	Liquid metal elastomer (BiInSn)	20.5671765 (Y- polarization)	92.0567177 (Y- polarization)	71.4895411	NO	NO	Shape change (stretching)
16	S14	Liquid metal elastomer (Ga)	92.0567177 (Y- polarization)	99.8004738 (Y- polarization)	7.7437561	NO	NO	Shape change (stretching)
17	S14	Liquid metal foamed elastomer (Ga)	99.9999000 (Y- polarization)	99.9999997 (Y- polarization)	0.0000997	NO	NO	Shape change (stretching)
18	S15	Ag NPs/SEBS	99.8415107 (Y- polarization)	99.9996838 (Y- polarization)	0.1581731	NO	NO	Shape change (stretching)
19	S16	CNT/TPU	94.3765867 (Y- polarization)	99.9653263 (Y- polarization)	5.5887396	NO	NO	Shape change (stretching)
20	S17	Liquid metal GaIn24.5/Ni	99.9999000 (Y- polarization)	99.9999900 (Y- polarization)	0.0000900	NO	NO	Shape change (stretching)
21	S18	PEDOT:PSS/ waterborne PU	99.9996019 (Y- polarization)	99.9998415 (Y- polarization)	0.0002396	NO	NO	Shape change (stretching)
22	S19	VO ₂ /CNF	98.6510371 (Y- polarization)	99.9994752 (Y- polarization)	1.3484381	NO	NO	Temperature change
23	S20	VO ₂ /EPM foam	5.59391237 (Y- polarization)	52.6848741 (Y- polarization)	47.0909617	NO	NO	Temperature change

24	S21	VO ₂ /PVDF-HFP	80.0473769 (Y- polarization)	99.9993393 (Y- polarization)	19.9519625	NO	NO	Temperature change
25	S22	Ti ₃ C ₂ T _x -WVO ₂	99.8711750 (Y- polarization)	99.9947519 (Y- polarization)	0.1235769	NO	NO	Temperature change
26	S23	RGO/VO ₂ -300°C	60.1892829 (Y- polarization)	92.0567177 (Y- polarization)	31.8674347	NO	NO	Temperature change
27	S23	RGO/VO ₂	84.1510681 (Y- polarization)	99.9996019 (Y- polarization)	15.8485338	NO	NO	Temperature change
28	S24	Graphene/PDMS	99.9910875 (Y- polarization)	99.9205672 (Y- polarization)	0.0705203	NO	NO	Temperature change
29	S25	Core-shell structural PNIPAM@p-PDA biomicrospheres	99.9601893 (Y- polarization)	99.99999997 (Y- polarization)	0.0398107	NO	NO	Humidity change
30	S26	Pyrolytic graphite-wet RGO/CNTs/PP non- woven spacer-pyrolytic graphite	97.4881136 (Y- polarization)	99.9841511 (Y- polarization)	2.4960375	NO	NO	Humidity change
31	S27	MXene film (Ti ₃ C ₂ T _x electrode in 1 M H ₂ SO ₄)/PET	99.9205672 (Y- polarization)	99.9521370 (Y- polarization)	0.0315698	NO	NO	Electroch- emical potential change
32	S27	MXene film (V ₂ CT _x electrode in 1 M H ₂ SO ₄)/PET	96.9800483 (Y- polarization)	99.2414224 (Y- polarization)	2.2613741	NO	NO	Electroch- emical potential change
33	S28	MXene/CNF aerogels	99.9800474 (X- polarization)	99.9999499 (Y- polarization)	0.0199025	YES	NO	Rotation angle change
34	S29	MXene@wood	99.8180299 (X- polarization)	99.9998262 (Y- polarization)	0.1817963	YES	NO	Rotation angle change
35	S30	Polyvinyl butyral/Ni- graphene/short-cut CF films	99.0000000 (X- polarization)	99.9205672 (Y- polarization)	0.9205672	YES	NO	Rotation angle change
36	S31	CF reinforced polymer	90.0000000 (X- polarization)	99.9987411 (Y- polarization)	9.9987411	YES	NO	Rotation angle change

37	S32	CF hybrid fabrics	43.7658675 (X- polarization)	80.0473769 (Y- polarization)	36.2815094	NO	NO	Rotation angle change
38	S33	RGO@directional porous carbon	99.6764063 (X- polarization)	99.9597283 (Y- polarization)	0.2833220	YES	NO	Rotation angle change
39	S34	CNT/NFC	99.9553316 (X- polarization)	99.9999937 (Y- polarization)	0.0446620	YES	NO	Rotation angle change
40	S35	Co based amorphous wires	12.9036410 (X- polarization)	99.6611558 (Y- polarization)	86.7575148	NO	NO	Rotation angle change
41	S36	MXene@PDA	36.9042656 (X- polarization)	98.7410746 (Y- polarization)	61.8368090	NO	NO	Rotation angle change
42	S37	Balsa tangential-section carbonized wood	99.9000000 (X- polarization)	99.9999990 (Y- polarization)	0.0999990	YES	NO	Rotation angle change
43	S38	Ordered RGO fiber film	68.3772234 (X- polarization)	99.9205672 (Y- polarization)	31.5433438	NO	NO	Rotation angle change
44	S39	CNT-based spacer fabric with CVD	99.9498813 (X- polarization)	99.9999998 (Y- polarization)	0.0501185	YES	NO	Rotation angle change
45	This work	GAFM	3.1722144 (X- polarization)	99.7852170 (Y- polarization)	96.6130026	YES (+ ⊥ CF)	YES (+ // CF)	Rotation angle change

Note: nanoparticles (NP), trans-1,4-polyisoprene (TPI), ethylene–vinyl acetate copolymer (EVA), polypyrrole (PPy), silver (Ag), melamine foam (MF), nanowire (NW), poly(ethylene glycol) (PEG), wheat flour (WF), carbon nanotube (CNT), polyurethane (PU), poly-dopamine (PDA), polydimethylsiloxane (PDMS), reduced graphene oxide (RGO), 3 dimension (3D), styrene-(ethylenebutylene)-styrene (SEBS), thermoplastic polyurethane (TPU), cellulose nanofibrils (CNF), poly(3,4-ethylenedioxythiophene)/poly(styrenesulfonate) (PEDOT:PSS), carbon fiber (CF), vanadium dioxide (VO₂), expanded polymer microsphere (EPM), poly (vinylidene fluoride-co-hexafluoropropylene (PVDF-HFP), poly(N-isopropylacrylamide)@porous polydopamine (PNIPAM@p-PDA), polypropylene (PP), poly(ethylene terephthalate) (PET), nano-brillated cellulose (NFC), chemical vapor deposition (CVD). Full polarization shielding refers to the EMI shielding efficiency being higher than 90% in both the X and Y polarization.”

References:

- [S1] Liu, L., *et al.* Off/on switchable smart electromagnetic interference shielding aerogel. *Matter* **4**, 1735–1747 (2021).
- [S2] Jia, X., Shen, B., Zhang, L., Zheng, W. Construction of shape-memory carbon foam composites for adjustable EMI shielding under self-fixable mechanical deformation. *Chem. Eng. J.* **405**, 126927 (2021).
- [S3] Chen, J., *et al.* Multifunctional shape memory foam composites integrated with tunable electromagnetic interference shielding and sensing. *Chem. Eng. J.* **466**, 143373 (2023).
- [S4] He, Y., *et al.* Multifunctional phase change composites based on elastic MXene/silver nanowire sponges for excellent thermal/solar/electric energy storage, shape memory, and adjustable electromagnetic interference shielding functions. *ACS Appl. Mater. Interfaces* **14**, 6057–6070 (2022).
- [S5] Chen, Y., Liu, Y., Li, Y., Qi, H. Highly sensitive, flexible, stable, and hydrophobic biofoam based on wheat flour for multifunctional sensor and adjustable EMI shielding applications. *ACS Appl. Mater. Interfaces* **13**, 30020–30029 (2021).
- [S6] Wang, G., *et al.* Structural design of compressible shape-memory foams for smart self-fixable electromagnetic shielding with reduced reflection. *Mater. Today Phys.* **22**, 100612 (2022).
- [S7] Wang, S., Wang, Z., Zheng, S. Y., Yang, J. Multifunctional heterostructured composite foam with tunable electromagnetic interference shielding. *Compos. Sci. Technol.* **248**, 110482 (2024).
- [S8] Li, J., *et al.* Oriented magnetic liquid metal-filled interlocked bilayer films as multifunctional smart electromagnetic devices. *Nano Res.* **16**, 1764–1772 (2023).
- [S9] Shen, B., Li, Y., Zhai, W., Zheng, W. Compressible graphene-coated polymer foams with ultralow density for adjustable electromagnetic interference (EMI) shielding. *ACS Appl. Mater. Interfaces* **8**, 8050–8057 (2016).
- [S10] Bian, J., *et al.* High-strain-sensitive dynamically adjustable electromagnetic interference shielding elastomer with pre-linked nickel chains. *Sci. China Mater.* **67**, 629–641 (2024).
- [S11] Guo, H., *et al.* Tough, stretchable dual-network liquid metal-based hydrogel toward high-performance intelligent on-off electromagnetic interference shielding, human motion detection and self-powered application. *Nano Energy* **114**, 108678 (2023).
- [S12] Zhu, R., *et al.* Anisotropic magnetic liquid metal film for wearable wireless electromagnetic sensing and smart electromagnetic interference shielding. *Nano Energy* **92**, 106700 (2022).
- [S13] Yao, B., *et al.* Highly stretchable polymer composite with strain-enhanced electromagnetic

- interference shielding effectiveness. *Adv. Mater.* **32**, 1907499 (2020).
- [S14] Yu, D., *et al.* A super-stretchable liquid metal foamed elastomer for tunable control of electromagnetic waves and thermal transport. *Adv. Sci.* **7**, 2000177 (2020).
- [S15] Liu, Z., *et al.* A general approach for buckled bulk composites by combined biaxial stretch and layer-by-layer deposition and their electrical and electromagnetic applications. *Adv. Electron. Mater.* **5**, 1800817 (2019).
- [S16] Feng, D., Xu, D., Wang, Q., Liu, P. Highly stretchable electromagnetic interference (EMI) shielding segregated polyurethane/carbon nanotube composites fabricated by microwave selective sintering. *J. Mater. Chem. C* **7**, 7938–7946 (2019).
- [S17] Zhang, M., *et al.* Stretchable liquid metal electromagnetic interference shielding coating materials with superior effectiveness. *J. Mater. Chem. C* **7**, 10331–10337 (2019).
- [S18] Li, P., Du, D., Guo, L., Guo, Y., Ouyang, J. Stretchable and conductive polymer films for high-performance electromagnetic interference shielding. *J. Mater. Chem. C* **4**, 6525–6532 (2016).
- [S19] Liao, S.-Y., *et al.* Intelligent shielding material based on VO₂ with tunable near-field and far-field electromagnetic response. *Chem. Eng. J.* **464**, 142596 (2023).
- [S20] Liao, S., *et al.* Reversible switching between microwave absorption and EMI shielding of VO₂ composite foam. *Small* **20**, 2402841 (2024).
- [S21] Liang, S., *et al.* Tunable high-performance electromagnetic interference shielding of VO₂ nanowires-based composite. *ACS Appl. Mater. Interfaces* **16**, 21024–21033 (2024).
- [S22] Qian, H., *et al.* Pushing electromagnetic interference shielding self-enhanced based on smart Ti₃C₂T_x-WVO₂ thermal management composite. *Carbon* **210**, 118081 (2023).
- [S23] Cheng, Z., *et al.* Intelligent off/on switchable microwave absorption performance of reduced graphene oxide/VO₂ composite aerogel. *Adv. Funct. Mater.* **32**, 2205160 (2022).
- [S24] Gao, W., *et al.* High-efficiency electromagnetic interference shielding realized in nacre-mimetic graphene/polymer composite with extremely low graphene loading. *Carbon* **157**, 570–577 (2020).
- [S25] Li, C., *et al.* Succulent-inspired implicit structural change for smart “ON/OFF” switchable and flexible EMI shielding coating. *ACS Appl. Mater. Interfaces* **16**, 12939–12950 (2024).
- [S26] Wang, Y., *et al.* Hydro-sensitive sandwich structures for self-tunable smart electromagnetic shielding. *Chem. Eng. J.* **344**, 342–352 (2018).
- [S27] Han, M., *et al.* Electrochemically modulated interaction of MXenes with microwaves. *Nat. Nanotechnol.* **18**, 373–379 (2023).
- [S28] Zeng, Z., *et al.* Nanocellulose-MXene biomimetic aerogels with orientation-tunable electromagnetic interference shielding performance. *Adv. Sci.* **7**, 2000979 (2020).

- [S29] Wei, Y., *et al.* Highly anisotropic MXene@wood composites for tunable electromagnetic interference shielding. *Compos. Part A Appl. Sci. Manuf.* **168**, 107476 (2023).
- [S30] Wen, B., Wang, X., Zhang, Y. Ultrathin and anisotropic polyvinyl butyral/Ni-graphite/short-cut carbon fibre film with high electromagnetic shielding performance. *Compos. Sci. Technol.* **169**, 127–134 (2019).
- [S31] Hong, J., Xu, P. Electromagnetic interference shielding anisotropy of unidirectional CFRP composites. *Materials.* **14**, 1907 (2021).
- [S32] Hong, X., *et al.* Polarization selection characteristics of carbon fiber orientation and interweaving for electromagnetic interference shielding behaviors. *Text. Res. J.* **92**, 269–283 (2022).
- [S33] Liu, Z., *et al.* Gradient in-plane oriented porous carbon inspired by fabrication of toasts for elegant EMI shielding performance. *Carbon* **207**, 136–143 (2023).
- [S34] Zeng, Z., *et al.* Nanocellulose assisted preparation of ambient dried, large-scale and mechanically robust carbon nanotube foams for electromagnetic interference shielding. *J. Mater. Chem. A* **8**, 17969–17979 (2020).
- [S35] Dai, X., *et al.* A smart amorphous wire composite with tunable electromagnetic shielding. *Small Struct.* **5**, 2300405 (2024).
- [S36] Deng, Z., *et al.* Controllable surface-grafted MXene inks for electromagnetic wave modulation and infrared anti-counterfeiting applications. *ACS Nano* **16**, 16976–16986 (2022).
- [S37] Dai, Z., *et al.* Highly anisotropic carbonized wood as electronic materials for electromagnetic interference shielding and thermal management. *Adv. Electron. Mater.* **9**, 2300162 (2023).
- [S38] Xu, L., *et al.* Ultrathin, ultralight, and anisotropic ordered reduced graphene oxide fiber electromagnetic interference shielding membrane. *Adv. Mater. Technol.* **6**, 2100531 (2021).
- [S39] Li, D., *et al.* 3D-structured carbon nanotube fibers as ultra-robust fabrics for adaptive electromagnetic shielding. *Nano Res.* **17**, 8521–8530 (2024).”

Response to Reviewer #3

Original comment:

The authors have addressed the concerns and therefore, the paper can be accepted in its present form.

REPLY:

We sincerely thank you for your help. Your suggestions have greatly improved our manuscript. We wish you all the best!

Response to Reviewer #4

Original comment:

In my opinion, the authors have adequately addressed referees' comments. Therefore, it can be accepted in the present form.

REPLY:

We sincerely appreciate your support and recognition of our article. We wish you all the best!